# The imprinted *Zdbf2* gene finely tunes control of feeding and growth in neonates

**Juliane Glaser**[1†], **Julian Iranzo**[1], **Maud Borensztein**[1], **Mattia Marinucci**[1], **Angelica Gualtieri**[2], **Colin Jouhanneau**[3], **Aurélie Teissandier**[1], **Carles Gaston-Massuet**[2], **Deborah Bourc'his**[1]*

[1]Institut Curie, PSL Research University, INSERM, CNRS, Paris, France; [2]Centre for Endocrinology, William Harvey Research Institute, Barts and the London School of Medicine and Dentistry, Queen Mary University of London, London, United Kingdom; [3]Institut Curie, PSL Research University, Animal Transgenesis Platform, Paris, France

**Abstract** Genomic imprinting refers to the mono-allelic and parent-specific expression of a subset of genes. While long recognized for their role in embryonic development, imprinted genes have recently emerged as important modulators of postnatal physiology, notably through hypothalamus-driven functions. Here, using mouse models of loss, gain and parental inversion of expression, we report that the paternally expressed *Zdbf2* gene controls neonatal growth in mice, in a dose-sensitive but parent-of-origin-independent manner. We further found that *Zdbf2*-KO neonates failed to fully activate hypothalamic circuits that stimulate appetite, and suffered milk deprivation and diminished circulating Insulin Growth Factor 1 (IGF-1). Consequently, only half of *Zdbf2*-KO pups survived the first days after birth and those surviving were smaller. This study demonstrates that precise imprinted gene dosage is essential for vital physiological functions at the transition from intra- to extra-uterine life, here the adaptation to oral feeding and optimized body weight gain.

*For correspondence:
deborah.bourchis@curie.fr

Present address: †RG Development & Disease, Max Planck Institute for Molecular Genetics, Berlin, Germany

## Editor's evaluation

The paper provides an elegant demonstration of the phenotypic consequences of genomic imprinting for postnatal physiology, focusing on the specific case of the Zdbf2 gene in the mouse. Using a series of gain and loss function models they explore how manipulating gene dosage independently of the parent of origin, influences the hypothalamic-pituitary endocrine axis, to regulate feeding behavior and growth during the early post-natal period.

## Introduction

Genomic imprinting is the process by which a subset of genes is expressed from only one copy in a manner determined by the parental origin. In mammals, genomic imprinting arises from sex-specific patterning of DNA methylation during gametogenesis, which generates thousands of germline differentially methylated regions (gDMRs) between the oocyte and the spermatozoa. After fertilization, the vast majority of gDMRs are lost during the epigenetic reprogramming that the embryonic genome undergoes (*Seah and Messerschmidt, 2018*). However, some gDMRs are protected through sequence- and DNA methylation-specific recruitment of the KRAB-associated protein 1 (KAP1) complex (*Li et al., 2008*; *Quenneville et al., 2011*; *Takahashi et al., 2019*) and become fixed as imprinting control regions (ICRs). Roughly 20 ICRs maintain parent-specific DNA methylation

throughout life and across all tissues in mouse and human genomes, and control the mono-allelic and parent-of-origin expression of approximately 150 imprinted genes (*Schulz et al., 2008*; *Tucci et al., 2019*).

DNA methylation-based genome-wide screens have led to the conclusion that all life-long ICRs have probably been discovered (*Proudhon et al., 2012*; *Xie et al., 2012*). However, a greater number of regions are subject to less robust forms of imprinted DNA methylation, restricted to early development or persisting in specific cell lineages only. Moreover, imprinted genes are often expressed in a tissue- or stage-specific manner (*Proudhon et al., 2012*; *Andergassen et al., 2017*; *Monteagudo-Sánchez et al., 2019*), adding to the spatio-temporal complexity of genomic imprinting regulation. Finally, while the vast majority of imprinted genes are conserved between mice and humans, a subset of them have acquired imprinting more recently in a species-specific manner (*Bogutz et al., 2019*). Why is reducing gene dosage important for imprinted genes, why does it occur in a parent-of-origin manner and why is it essential for specific organs in specific species are fundamental questions in mammalian development and physiology.

Imprinted genes have long-recognized roles in development and viability in utero, by balancing growth and resource exchanges between the placenta and the fetus. Moreover, it is increasingly clear that imprinted genes also strongly influence postnatal physiology (*Peters, 2014*). Neonatal growth, feeding behavior, metabolic rate, and body temperature are affected by improper dosage of imprinted genes in mouse models and human imprinting disorders (*Charalambous et al., 2014*; *Ferrón et al., 2011*; *Leighton et al., 1995*; *Li et al., 1999*; *Plagge et al., 2004*; *Nicholls et al., 1989*; *Buiting, 2010*). Imprinting-related postnatal effects are recurrently linked to dysfunction of the hypothalamus (*Ivanova and Kelsey, 2011*), a key organ for orchestrating whole body homeostasis through a complex network of nuclei that produce and deliver neuropeptides to distinct targets, including the pituitary gland that in turn secretes endocrine hormones such as the growth hormone (GH). Accordingly, the hypothalamus appears as a privileged site for imprinted gene expression (*Gregg et al., 2010*; *Higgs et al., 2021*). A typical illustration of such association is provided by a cluster of hypothalamic genes whose dosage is altered in Prader-Willi syndrome (PWS). PWS children present neurological and behavioral impairments in particular related to feeding, in the context of hypothalamic neuron anomalies (*Swaab, 1997*; *Cassidy and Driscoll, 2009*). In mouse models, single inactivation of the PWS-associated *Magel2* gene results in neonatal growth retardation, reduced food intake and altered metabolism (*Bischof et al., 2007*; *Kozlov et al., 2007*; *Schaller et al., 2010*). Fine-tuning hypothalamic inputs is particularly important for adapting to environmental changes, the most dramatic one for mammals being the transition from intra- to extra-uterine life at birth. Early mis-adaptation to postnatal life can have far-reaching consequences on adult health, by increasing the risk of metabolic diseases. It therefore is of the utmost importance to thoroughly document the action of imprinted genes, particularly in hypothalamic functions.

*Zdbf2* (*DBF-type zinc finger- containing protein 2*) is a paternally expressed gene with preferential expression in the brain (*Kobayashi et al., 2009*; *Greenberg et al., 2017*). It is one of the last-discovered genes with life-long and tissue-wide imprinted methylation, and conserved imprinting in mice and humans, however its function is not yet resolved (*Kobayashi et al., 2009*; *Duffié et al., 2014*). We previously characterized the complex parental regulation of the *Zdbf2* locus: it is controlled by a maternally methylated gDMR during the first week of embryogenesis, but for the rest of life, it harbors a somatic DMR (sDMR) that is paternally methylated (*Proudhon et al., 2012*; *Duffié et al., 2014*; *Figure 1A*). The maternal gDMR coincides with a promoter that drives paternal-specific expression of a *Long isoform of Zdbf2* (*Liz*) transcript. In the pluripotent embryo, *Liz* transcription triggers in cis DNA methylation at the sDMR, allowing de-repression of the canonical promoter of *Zdbf2*, located ~10 kb downstream (*Greenberg et al., 2017*; *Figure 1A*). In fact, although specifically expressed in the embryo where it undergoes stringent multi-layered transcriptional control (*Greenberg et al., 2019*), *Liz* is dispensable for embryogenesis itself. Its sole function seems to epigenetically program expression of *Zdbf2* later in life: genetic loss-of-function of *Liz* (*Liz*-LOF) prevents methylation of the sDMR, giving rise to mice that cannot activate *Zdbf2*–despite an intact genetic sequence–in the hypothalamus and the pituitary gland and display reduced postnatal growth (*Greenberg et al., 2017*). The cause of this growth phenotype and whether it is directly linked to the function and dosage regulation of *Zdbf2* in the brain cells is unknown.

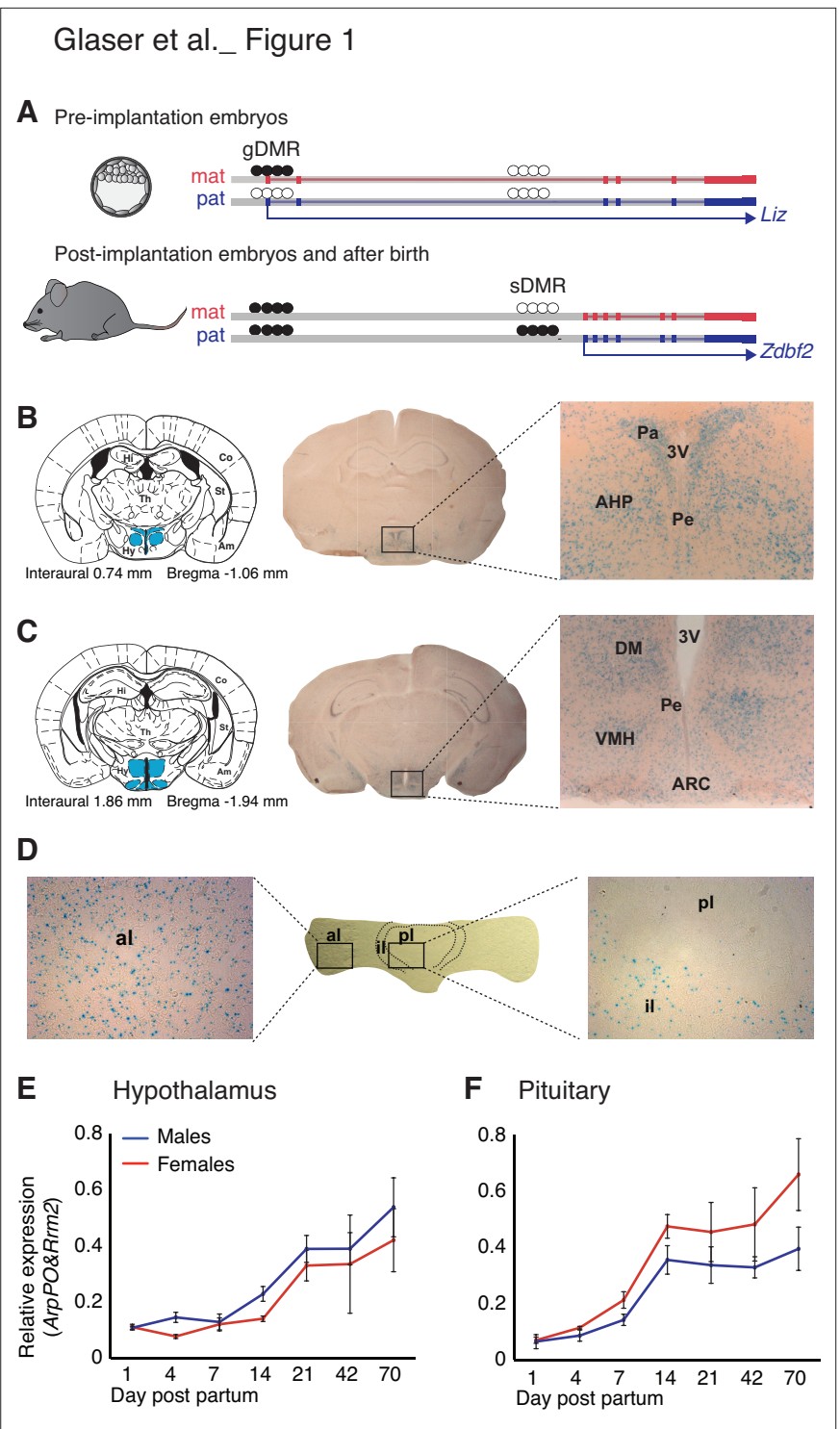

**Figure 1.** *Zdbf2* expression localizes preferentially in the neuro-endocrine cells of the hypothalamo-pituitary axis in juvenile animals. (**A**) Scheme of the *Liz/Zdbf2* locus regulation during mouse development. In the pre implantation embryos, a maternally methylated gDMR allow the paternal-specific expression of the *Long isoform of Zdbf2* (*Liz*). *Liz* expression triggers, in cis, DNA methylation at the sDMR which is localized 8 kb upstream of *Zdbf2* canonical promoter. In the post-implantation embryos and for the rest of the life, the imprint at the locus is controlled by the paternal methylation at the sDMR and this allow de-repression of *Zdbf2*, leading to its paternal-specific expression in the postnatal brain. (**B, C**) X-gal staining on brain coronal sections from 2-week-old *Zdbf2*-lacZ transgenic males. The coronal diagram from the Mouse Brain Atlas (left panel) localizes the sections in zone 40 (**B**) and zone

*Figure 1 continued*

47 (**C**) in stereotaxic coordinates, with the hypothalamus indicated in blue. Whole brain coronal sections (middle panel) show specific staining in the hypothalamus, due to several positive hypothalamic nuclei (right panel, 20 X magnificence). Hi, hippocampus; Co, cortex; Th, thalamus; Am, amygdala; St, striatum; Hy, hypothalamus; 3 V, third ventricle; Pa, paraventricular hypothalamic nucleus; Pe, periventricular hypothalamic nucleus; AH, anterior hypothalamic area; DM, dorsomedial hypothalamic nucleus; VMH, ventromedial hypothalamic nucleus; Arc, arcuate hypothalamic nucleus. (**D**) X-gal staining on pituitary horizontal sections. The posterior lobe of the pituitary shows no X-gal staining (right panel), while staining is evenly distributed in the anterior lobe (left panel). pl, posterior lobe; il, intermediate lobe; al, anterior lobe. (**E, F**) *Zdbf2* expression measured by RT-qPCR in the hypothalamus (**E**) and the pituitary (**F**) from 1 to 70 days after birth. Data are shown as mean ± s.e.m. of n = 3 C57Bl6/J mice.

The online version of this article includes the following figure supplement(s) for figure 1:

**Figure supplement 1.** *Zdbf2* expression from pituitary and hypothalamus.

By generating loss-of-function (LOF) and gain-of-function (GOF) mouse mutants, we show here that ZDBF2 is necessary for optimal growth and survival during the nursing period, by stimulating hypothalamic food circuits immediately at birth. Moreover, our data support that the dose but not the parental origin of *Zdbf2* expression is important for its imprinted mode of action. Altogether, our study illustrates the critical function and proper dose regulation of *Zdbf2* for adaptation to postnatal life.

## Results
### *Zdbf2* is expressed in the neuro-endocrine cells of the hypothalamo-pituitary axis

Besides a C2H2 zinc finger motif, the ZDBF2 protein does not contain any obvious functional domain that could inform its molecular function. To gain insights into the role of *Zdbf2*, we first examined the cellular specificity and temporal dynamics of its expression. Comprehensive datasets of adult tissues expression in mice and human suggested *Zdbf2* expression is prevalent in brain tissues and pituitary gland (biogps in mouse tissues: http://biogps.org/#goto=genereport&id=73884 and GTex in human tissues: https://gtexportal.org/home/gene/ZDBF2). Our data endorse this brain and pituitary-specific expression of *Zdbf2* in adult mice and show the highest level of expression in the hypothalamus (*Figure 1—figure supplement 1A*; *Greenberg et al., 2017*). Similarly, *Zdbf2* expression was predominantly observed in the brain and the spinal cord of embryos (http://www.emouseatlas.org), a feature we confirmed at embryonic day E12.5 using a previously described *LacZ* reporter *Zdbf2* gene-trap mouse line (*Figure 1—figure supplement 1B*; *Greenberg et al., 2017*).

Using this same *Zdbf2*-LacZ reporter line, we confirmed the expression of *Zdbf2* in various brain tissues at 2 weeks of age, and more specifically, the high specificity in hypothalamic cells that belonged to the peri- and paraventricular nuclei in the anterior area (*Figure 1B*), and the arcuate, the dorsomedial and the ventromedial nuclei in the lateral hypothalamic area (*Figure 1C*). Analysis of publicly available single-cell RNA-sequencing (scRNA-seq) data further defined *Zdbf2* expression to be specific to neuronal cells and absent from non-neuronal cells of the hypothalamus (*Figure 1—figure supplement 1C-D*; *Chen et al., 2017*). The three hypothalamic clusters where *Zdbf2* was the most highly expressed were both glutamatergic (Glu13 and Glu15) and GABAergic (GABA17) neurons from the arcuate and the periventricular hypothalamic regions (*Figure 1—figure supplement 1C*). Same results were recently reported using different scRNA-seq datasets (*Higgs et al., 2021*). Interestingly, these nuclei synthesize peptides that stimulate hormone production from the pituitary gland, or that control energy balance–food intake, energy expenditure, and body temperature–by directly acting on the brain and/or more distal organs (*Saper and Lowell, 2014*).

*Zdbf2* being also expressed in the pituitary gland (*Figure 1—figure supplement 1A*), we hypothesized it could have a role in the hypothalamo-pituitary axis. When we examined the pituitary gland by X-gal staining, we found *Zdbf2* to be expressed in the anterior and intermediate lobes of the gland and almost undetectable signal in the posterior lobe (*Figure 1D* and *Figure 1—figure supplement 1E*), a pattern that was confirmed from available scRNA-seq data (*Figure 1—figure supplement 1F*; *Cheung et al., 2018*). The anterior and intermediate lobes that form the adenohypophysis are

responsible for hormone production (*Mollard et al., 2012*). The anterior lobe secretes hormones from five different specialized hormone-producing cells under the control of hypothalamic inputs (growth hormone-GH, adrenocorticotropic hormone-ACTH, thyroid stimulating hormone-TSH, luteinizing hormone-LH and prolactin-PRL) and the intermediate lobe contains melanotrope cells (*Kelberman et al., 2009*). Altogether, the expression specificity of *Zdbf2* suggests a role in functions of the endocrine hypothalamo-pituitary axis and/or of the hypothalamus alone. Finally, we further found that steady-state levels of *Zdbf2* transcripts progressively rose in the hypothalamus and the pituitary gland after birth, reached their maximum at 2–3 weeks and then stabilized at later ages, in both males and females (*Figure 1E and F*). The expression of *Zdbf2* therefore mostly increases in juvenile pups prior to weaning.

## ZDBF2 positively regulates growth postnatally

In *Liz*-LOF mutants, *Zdbf2* failed to be activated and animals displayed postnatal body weight reduction (*Greenberg et al., 2017*). However, whether this was directly and only linked to *Zdbf2* deficiency was not resolved. To directly probe the biological role of ZDBF2, we therefore generated a mouse model of a genetic loss-of function of *Zdbf2*. More specifically, we engineered a ~ 700 bp deletion of the entirety of exon 6 (*Figure 2A*) that is common to all annotated *Zdbf2* transcripts (*Duffié et al., 2014*). The *Zdbf2*-Δexon6 deletion induces a frame-shift predicted to generate a severely truncated protein (wild-type 2494aa versus 19aa mutant protein) that notably lacks the zinc finger motif. As *Zdbf2* is an imprinted gene with paternal-specific expression, the deletion should exhibit an effect upon paternal but not maternal transmission. For simplicity, heterozygous mutants with a paternally inherited *Zdbf2*-Δexon6 deletion are thus referred to as *Zdbf2*-KO thereafter.

At birth, we found that *Zdbf2*-KO animals were present at expected sex and Mendelian ratios (*Figure 2—figure supplement 1A-B*). However, while there was no difference in body mass prior to birth (E18.5), *Zdbf2*-KO neonates of both sexes appeared smaller than WT littermates already at 1 and 5 days of postnatal life (day post-partum, dpp) (*Figure 2B–C*), indicating slower growth in the first days after birth. At 15dpp, *Zdbf2*-KO mice were 20% lighter than their WT littermates (*Figure 2B and E–F*). Lower body mass persisted into adulthood, measured up to 12 weeks of age (*Figure 2D*). As predicted, when present on the maternal allele, the *Zdbf2*-Δexon six deletion had no discernable growth effect (*Figure 2—figure supplement 1C-D*).

The reduced body weight phenotype was highly penetrant (*Figure 2—figure supplement 1E*) and was not obviously due to a developmental delay: *Zdbf2*-KO pups and their WT littermates synchronously acquired typical hallmarks of postnatal development (skin pigmentation, hair appearance and eye opening) (*Figure 2—figure supplement 1F-I*). The growth phenotype appeared to be systemic: it affected both the body length and mass of *Zdbf2*-KO animals (*Figure 2—figure supplement 2A*) and a range of organs uniformly (*Figure 2—figure supplement 2B*). To gain insight into the origin of the body mass restriction, we performed dual-energy X-ray absorptiometry (DEXA) scan at 7 weeks of age to measure in vivo the volume fraction of the three dominant contributors to body composition: adipose, lean, and skeletal tissues (*Chen et al., 2012*). While we confirmed that adult *Zdbf2*-KO males exhibit an overall body mass reduction (*Figure 2G*), we did not observe difference in the composition of fat and bone tissues (*Figure 2H* and *Figure 2—figure supplement 2C-E*). However, *Zdbf2*-KO mice had decreased lean mass (*Figure 2I*). Collectively, the above analyses demonstrate that the product of *Zdbf2* positively regulates body mass gain in juvenile mice.

In conclusion, *Zdbf2*-KO animals exhibit the same growth defect that we previously reported in *Liz*-LOF mice, with identical postnatal onset and severity (*Greenberg et al., 2017*). However, contrary to the *Liz*-LOF mice, the regulatory landscape of the paternal allele of *Zdbf2* was intact in *Zdbf2*-KO mice: DNA methylation of the sDMR located upstream of *Zdbf2* was normal, and accordingly, *Zdbf2* transcription was activated (*Figure 2—figure supplement 2F-G*). This supports that *Liz* transcription in the early embryo—which is required for sDMR DNA methylation (*Greenberg et al., 2017*)—is not impacted in the *Zdbf2*-KO model. We therefore show that a genetic mutation of *Zdbf2* (*Zdbf2*-KO) and a failure to epigenetically program *Zdbf2* expression (*Liz*-LOF) are phenotypically indistinguishable. As *Liz*-LOF mice show a complete lack of *Zdbf2* expression (*Greenberg et al., 2017*), this incidentally supports that *Zdbf2*-KO mice carry a null *Zdbf2* allele. Unfortunately, we failed to specifically detect the mouse ZDBF2 protein with commercial or custom-made antibodies.

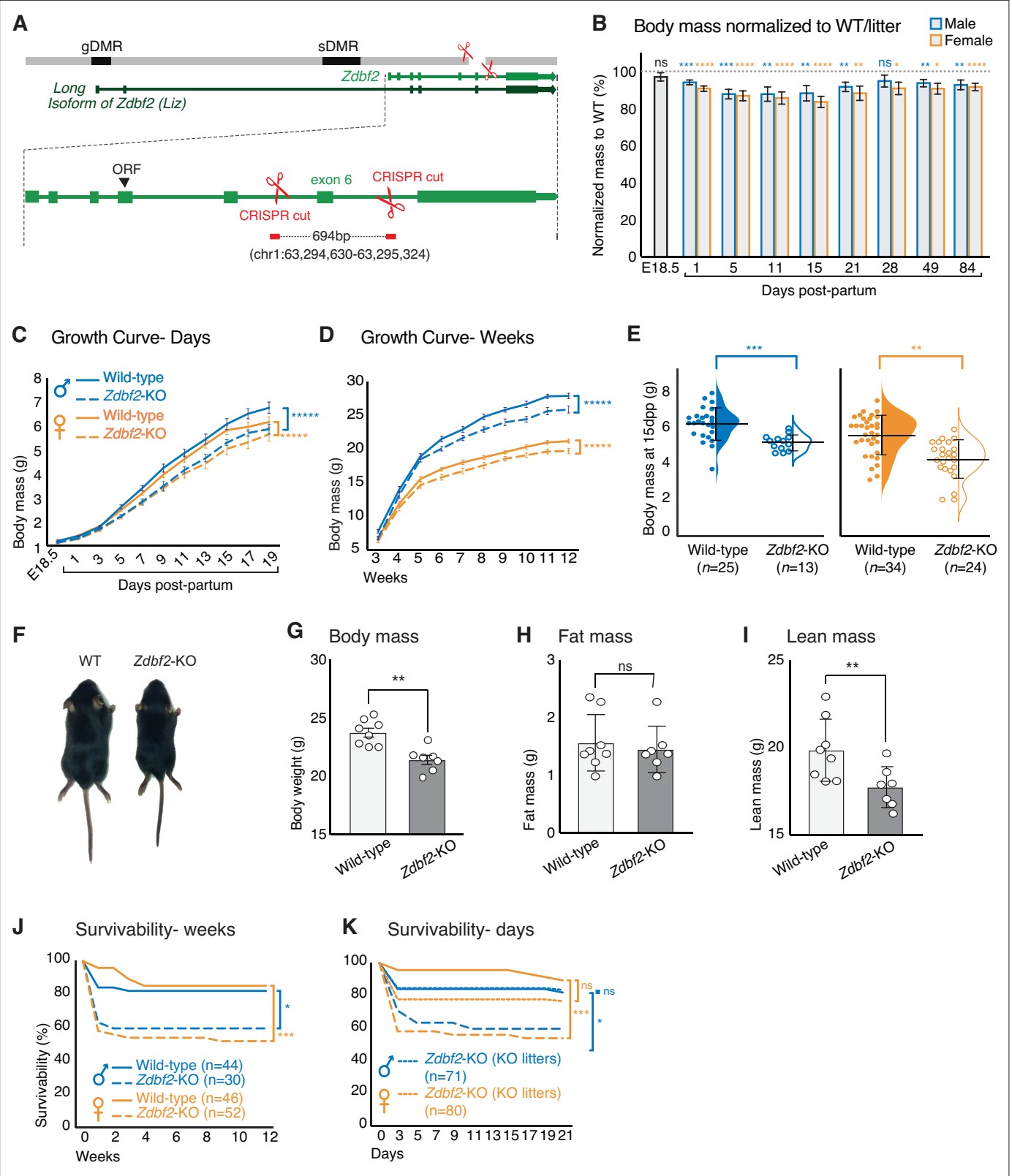

**Figure 2.** *Zdbf2*-KO mice exhibit growth reduction and partial postnatal lethality. (**A**) Graphical model of the *Zdbf2* deletion generated using two sgRNAs across exon 6. The two differentially methylated (DMR) regions of the locus are indicated (germline-gDMR, and somatic-sDMR), as well as the *Long Isoform of Zdbf2* (*Liz*). The ORF (open-reading frame) of *Zdbf2* starts in exon 4. Genomic coordinates of the deletion are indicated. (**B**) Body weight of *Zdbf2*-KO mice normalized to WT littermates (100%) followed from embryonic day E18.5–84 days post-partum. Data are shown as means ±

*Figure 2 continued on next page*

*Figure 2 continued*

s.e.m. from individuals from n = 27 litters. Statistical analyses were performed by a two-tailed, unpaired, non-parametric Mann Whitney t-test. **** p ≤ 0.0001, ***p ≤ 0.001,**p ≤ 0.01, *p ≤ 0.05. (C, D) Growth curve comparing the body weights of WT and *Zdbf2*-KO mice prior to weaning, from E18.5 to 19dpp (C) and over 3 months after birth (D). n = 15–50 mice were analyzed per genotype, depending on age and sex. Statistical analyses were performed by a two-way ANOVA test. **** p ≤ 0.0001. (E) Half dot plot- half violin plot showing the weight distribution in 2-week-old males (left) and females (right) of WT and *Zdbf2*-KO genotypes. Statistical analyses were performed by a two-tailed, unpaired, nonparametric Mann Whitney t test. ***p ≤ 0.001, **p ≤ 0.01. (F) Representative photography of a smaller 2 week-old *Zdbf2*-KO male compared to a WT littermate. (G–I) Dual-energy X-ray absorptiometry (DXA) analysis showing the calculation of body mass (G), fat mass (H) and lean mass (I) in WT and *Zdbf2*-KO males at 7 weeks. Data are shown as means ± s.e.m. from n = 8 WT and n = 7 *Zdbf2*-KO males. Statistical analyses were performed by a two-tailed, unpaired, nonparametric Mann Whitney t test. **p ≤ 0.01. (J–K) Kaplan-Meier curves of the survivability from birth to 12 weeks of age comparing WT (plain lines) and *Zdbf2*-KO (dotted lines) littermates from WT x *Zdbf2* KO/WT backcrosses (J). Impaired survivability occurs specifically from the first day to 2 weeks of age. Kaplan-Meier curves of the survivability from 1 to 21 dpp comparing *Zdbf2*-KO pups generated from WT x *Zdbf2* KO/KO intercrosses, with *Zdbf2*-KO and their WT littermates generated from WT x *Zdbf2* KO/WT backcrosses as in J (K). *Zdbf2*-KO pups are more prone to die only when they are in competition with WT littermates (small dotted lines), while *Zdbf2*-KO pups have a normal survivability when they are not with WT littermates (large dotted lines). Statistical analyses were performed by a two-tailed, Chi2 test on the last time point for each curves (12 weeks for (J) and 21 days for (K)). ***p ≤ 0.001, *p ≤ 0.05.

The online version of this article includes the following source data and figure supplement(s) for figure 2:

**Source data 1.** Survivability counts from different transmission of the *Zdbf2*-KO allele.

**Figure supplement 1.** *Zdbf2*-KO pups acquire normal hallmarks of postnatal development.

**Figure supplement 2.** Phenotypic and molecular characterization of *Zdbf2* mutants.

## *Zdbf2*-KO neonates are small or die prematurely within the first week of age

Growth restriction during the nursing period could dramatically impact the viability of *Zdbf2*-KO pups. Indeed, analysis of *Zdbf2*-KO cohorts revealed that although there was no Mendelian bias at 1dpp (*Figure 2—figure supplement 1B*), a strong bias was apparent at 20dpp: 75 WT and 44 *Zdbf2*-KO animals were weaned among n = 27 litters, while a 50/50 ratio was expected. Postnatal viability was the most strongly impaired within the first days after birth, with only 56% of *Zdbf2*-KO males and 52% of *Zdbf2*-KO females still alive at 3dpp, compared to 84% and 88% of WT survival at this age (*Figure 2J* and *Figure 2—source data 1A*). Importantly, a partial postnatal lethality phenotype was also present in *Liz*-LOF mutants who are equally growth-restricted as a result of *Zdbf2* deficiency (*Figure 2—figure supplement 1H* and *Figure 2—source data 1B*), but did not occur when the mutation was transmitted from the silent maternal allele harboring either the *Zdbf2* deletion (*Figure 2—figure supplement 1I* and *Figure 2—source data 1C*) or the *Liz* deletion (*Figure 2—figure supplement 1J* and *Figure 2—source data 1D*).

To determine whether partial lethality was due to a vital function of ZDBF2, per se, or to a competition with WT littermates, we monitored postnatal viability in litters of only *Zdbf2*-KO pups generated from crosses between WT females x homozygous *Zdbf2*-KO/KO males. Litter sizes were similar than the ones sired by *Zdbf2*-KO/WT males. However, when placed in an environment without WT littermates, *Zdbf2*-KO pups gained normal viability (*Figure 2K* and *Figure 2—source data 1E*). Perinatal mortality was thus rescued when there was no competition with WT littermates. Incidentally, this shows that *Zdbf2*-KO pups are mechanically able to ingest milk. Overall, our detailed study of the *Zdbf2*-KO phenotype reveals that smaller *Zdbf2*-KO neonates have reduced fitness compared to their WT littermates. In absence of ZDBF2, after a week of life, half of juvenile pups did not survive and the other half failed to thrive.

## Postnatal body weight control is highly sensitive to *Zdbf2* dosage

The physiological effects of imprinted genes are intrinsically sensitive to both decreases and increases in the expression of imprinted genes. We therefore went on assessing how postnatal body weight gain responded to varying doses of the imprinted *Zdbf2* gene. We first analyzed the growth phenotype resulting from partial reduction in *Zdbf2* dosage, as compared to a total loss of *Zdbf2* in *Zdbf2*-KO mice. For this, we used the *LacZ* reporter *Zdbf2* gene-trap mouse line (*Greenberg et al., 2017*), in which the insertion of the *LacZ* cassette downstream of exon five does not fully abrogate the production of full-length *Zdbf2* transcripts (*Figure 3—figure supplement 1A*). Upon paternal inheritance of this allele, animals still express 50% of WT full-length *Zdbf2* mRNA levels in the hypothalamus and the pituitary gland (*Figure 3—figure supplement 1B-C*). Interestingly, these mice displayed significant

weight reduction at 7 and 14dpp compared to their WT littermates (*Figure 3—figure supplement 1D*), but less severe than *Zdbf2*-KO mice (*Figure 2B–D*). At 14dpp, we observed an 8% body mass reduction in males, which is incidentally half the growth reduction observed in *Zdbf2*-KO mice for the same age range.

We then assessed the consequences of increased *Zdbf2* dosage with the hypothesis that this would lead to excessive body mass gain after birth as opposed to reduced *Zdbf2* dosage. For this, we took advantage of a *Zdbf2* gain-of-function (GOF) line that we serendipitously obtained when generating *Liz* mutant mice (*Greenberg et al., 2017*). The *Zdbf2*-GOF lines carry smaller deletions than what was initially aimed for (924 bp and 768 bp instead of 1.7 kb), resulting from non-homologous end joining (NHEJ) repair of the cut induced by the left sgRNA only (*Figure 3—figure supplement 1E-F*). Both deletions removed only 5' portions of the gDMR and *Liz* exon 1, leaving some part of exon 1 intact, and induces an epigenetic 'paternalization' of the maternal allele of the locus. While sDMR methylation exclusively occurs on the paternal allele in WT embryos, animals that maternally inherit this partial deletion also acquired sDMR methylation on the maternal allele (*Figure 3A* and *Figure 3—figure supplement 1F-H*). Indeed, important, but not complete, sDMR DNA methylation was observed (65% on the maternal allele compared to 95% on the paternal allele, *Figure 3—figure supplement 1H*). This was associated with *Zdbf2* activation from the maternal allele, although slightly less than from the WT paternal allele (*Figure 3B–C*). As a consequence, *Zdbf2*-GOF animals exhibit bi-allelic *Zdbf2* expression, with a net 1.7-fold increase of *Zdbf2* levels in postnatal hypothalamus and pituitary gland, as compared to WT littermates that express *Zdbf2* mono-allelically, from the paternal allele only. In sum, we have generated mutant mice with loss-of-imprinting of the *Zdbf2* locus.

Importantly, we found that increased *Zdbf2* dosage was impactful for postnatal body mass, specifically in males (*Figure 3D–H*). From 10 to 21dpp, *Zdbf2*-GOF juvenile males were 10% heavier when normalized to the average WT siblings from the same litter (*Figure 3D*) but their viability was similar to that of WT controls (*Figure 3—figure supplement 2A-C*). Similar to growth restriction observed in *Zdbf2*-KO mutant mice, males *Zdbf2*-GOF remained heavier than WT littermates into adult life, as measured until 10 weeks of age (*Figure 3F*), and a range of organs were uniformly affected (*Figure 3—figure supplement 2D*). *Zdbf2*-GOF males were significantly overweighed at 15dpp (*Figure 3G–H*) and the body mass distribution of males *Zdbf2*-GOF showed most of them were larger than WT controls (*Figure 3—figure supplement 2E*). Although *Zdbf2* was overexpressed in both *Zdbf2*-GOF males and females, the male-specific overgrowth phenotype may imply stimulation of the phenotype by sex hormones, directly or indirectly. Comparatively, progenies from a paternal transmission of the deletion showed a growth progression similar to their WT littermates (*Figure 3—figure supplement 2F-G*). Overall, these data illustrate the importance of *Zdbf2* for the regulation of postnatal body weight gain at the onset of postnatal life. We therefore demonstrate that the *Zdbf2* imprinted gene encodes a genuine positive regulator of postnatal body weight gain in mice, with highly attuned dose-sensitive effects.

## *Zdbf2* regulates postnatal growth in a parent-of-origin independent manner

Imprinted genes with a paternal expression generally tend to enhance prenatal and postnatal growth (*Haig, 2000*). Consistently, we found that the paternally expressed *Zdbf2* gene promotes postnatal weight gain. Having determined the dosage effect of *Zdbf2*, we next wondered what role plays the paternal origin of *Zdbf2* expression on postnatal body weight. To invert *Zdbf2* parental expression, we intercrossed the *Liz*-LOF and *Zdbf2*-GOF lines, which consist in a maternalization of the paternal allele and a paternalization of the maternal allele of *Zdbf2*, respectively. By crossing an heterozygous *Zdbf2*-GOF female with an heterozygous *Liz*-LOF male (*Figure 4A*, left panel), four genotypes can segregate in the progeny, with littermates displaying various dosage and parent-of-origin expression of *Zdbf2* (*Figure 4A*): (1) wild-type animals with one dose of paternal *Zdbf2* expression, (2) animals with a paternal *Liz*-LOF allele and lack of *Zdbf2* expression (equivalent to a *Zdbf2*-loss-of-function, *Zdbf2*-LOF), (3) animals with a maternal *Zdbf2*-GOF allele and biallelic *Zdbf2* expression, and finally, (4) animals with combined maternal *Zdbf2*-GOF and paternal *Zdbf2*-LOF alleles and potentially, mono-allelic, reverted maternal expression of *Zdbf2* (*Figure 4A*, right panel).

The uniform strain background of the two lines did not allow us to use strain-specific sequence polymorphisms to distinguish the parental origin of *Zdbf2* regulation in the compound

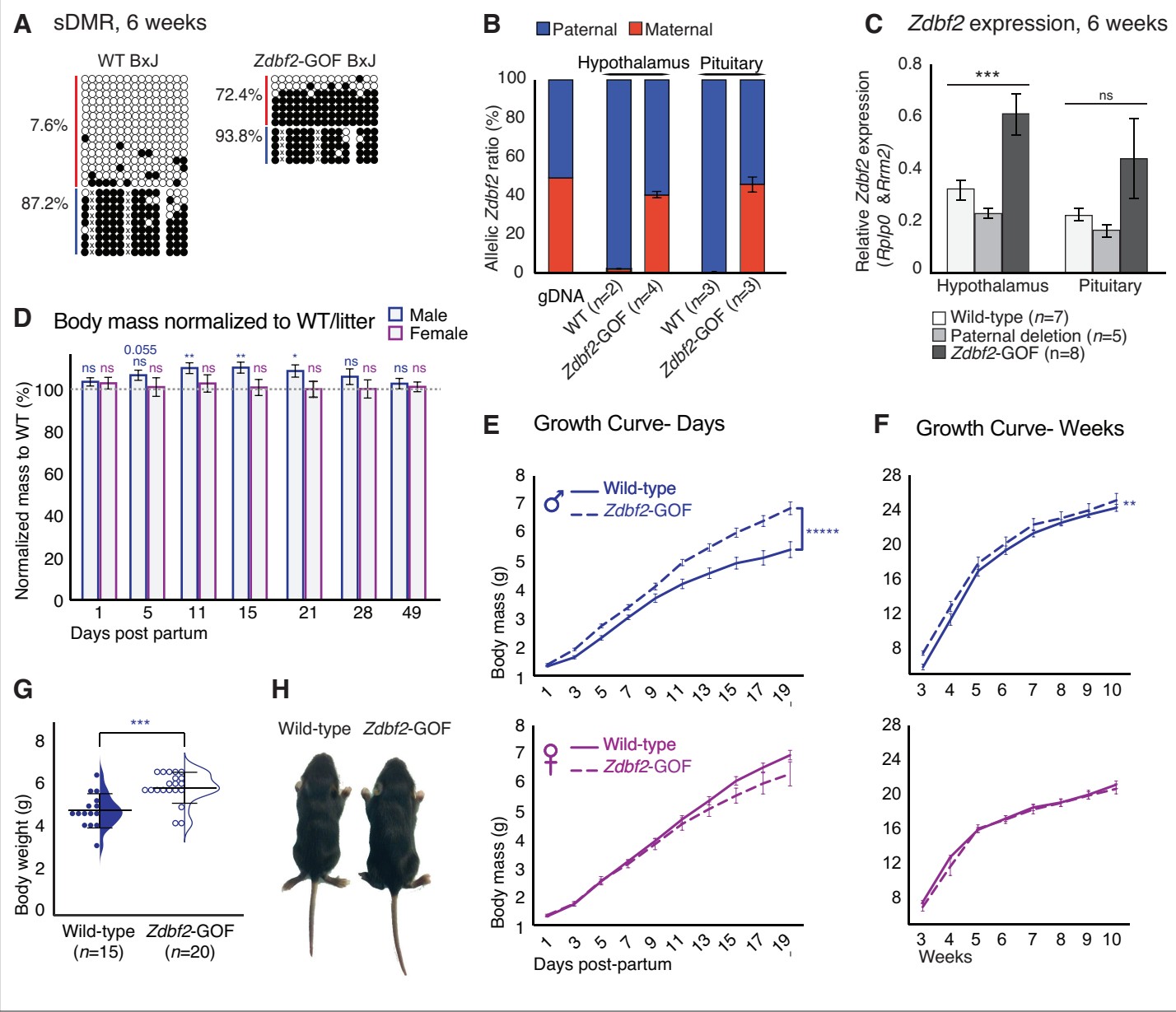

**Figure 3.** *Zdbf2* influences postnatal growth in a dose-dependent manner. (**A**) Bisulfite cloning and sequencing showing CpG methylation levels at the sDMR locus of hypothalamus DNA from 6-week-old hybrid WT (left) and *Zdbf2*-GOF (right) mice (*Zdbf2*-GOF± x JF1 cross). Red, maternal alleles; blue, paternal alleles. cross, informative JF1 SNP. (**B**) Allelic expression of *Zdbf2* in hypothalamus and pituitary gland from 3-week-old mice, measured by RT-pyrosequencing. Genomic DNA extracted from a C57Bl/6 x JF1 hybrid cross was used as a control for pyrosequencing bias. (**C**) RT-qPCR measurement reveals a ~ 1.7-fold-increase of *Zdbf2* expression in the hypothalamus and pituitary gland of 3-week-old mice with a maternal transmission of the deletion. Expression of *Zdbf2* in mice carrying the deletion on the paternal allele is similar to WT. Statistical analyses were performed by a one-way ANOVA test. ***$p \leq 0.001$. (**D**) Normalized body growth of *Zdbf2*-GOF mice to their WT littermates (100%) followed at different ages (1–84 days) from n = 14 litters. The overgrowth is seen specifically in males, from 5 to 28 days. Statistical analyses were performed by a two-tailed, unpaired, non-parametric Mann Whitney t-test. **$p \leq 0.01$, *$p \leq 0.05$. The number on top of the data at 5dpp indicate a non-significant but close to be p-value. (**E, F**) Growth curves of female and male mice, comparing the body weights of WT and *Zdbf2*-GOF, through the three first weeks of life (**D**) and through 10 weeks (**E**). n = 10–30 mice were analyzed per genotype, depending on age and sex. Statistical analyses were performed by a two-way ANOVA test. **** $p \leq 0.0001$, **$p \leq 0.01$. (**G**) Half dot- half violin plots showing the weight distribution at 2 weeks of age between WT and *Zdbf2*-GOF males. Data are shown as means ± s.e.m. from *n* individuals. Statistical analyses were performed by a two-tailed, unpaired, nonparametric Mann Whitney t test. *** $p \leq 0.005$. (**H**) Representative photography of a bigger *Zdbf2*-GOF male as compared to a WT littermate at 2 weeks.

The online version of this article includes the following figure supplement(s) for figure 3:

**Figure supplement 1.** Characterization of partial *Zdbf2*-LOF and *Zdbf2*-GOF mouse lines.

**Figure supplement 2.** Phenotypic characterization of *Zdbf2*-GOF mutants.

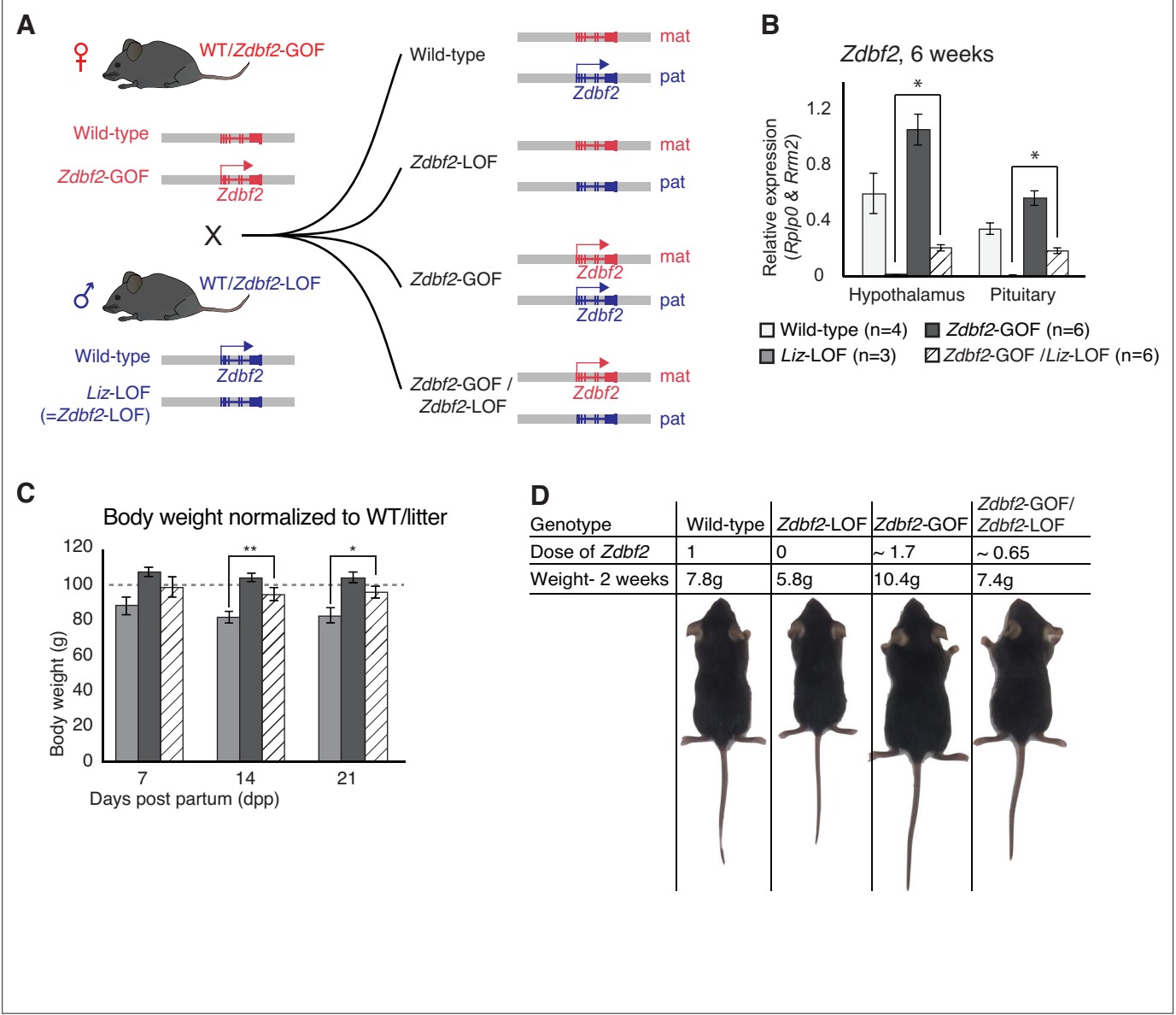

**Figure 4.** *Zdbf2* influences postnatal growth in a parent-of-origin-independent manner. (**A**) Scheme of the cross made to obtain embryos with an inversion of the parental origin of *Zdbf2* expression. *Zdbf2*-GOF heterozygote females were crossed with heterozygote males for the *Liz*-LOF deletion–which we demonstrated as equivalent to a *Zdbf2*-LOF allele–(left) to obtain one quarter of embryos expressing one dose of *Zdbf2* from the maternal allele (right, bottom). (**B**) *Zdbf2* expression in the hypothalamus and the pituitary gland is shown in males for each of the four possible genotypes. The level of *Zdbf2* in *Zdbf2*-GOF /*Zdbf2*-LOF mice almost completely rescues the defect seen in *Zdbf2*-LOF and *Zdbf2*-GOF mutants. Data are shown as means ± s.e.m. from *n* individuals. Statistical analyses were performed by a two-tailed, unpaired, nonparametric Mann Whitney t test. * p ≤ 0.05. (**C**) Normalized body growth of *Zdbf2*-LOF, *Zdbf2*-GOF and *Zdbf2*-GOF /*Zdbf2*-LOF males to their WT littermates (100%) followed at 7, 14, and 21 days after birth. *Zdbf2*-GOF /*Zdbf2*-LOF adult mice exhibit a body weight similar to the WT showing a partial rescue of the growth reduction and overgrowth phenotype due to respectively the lack of *Zdbf2* and the gain of *Zdbf2* expression in the brain. Data are shown as means ± s.e.m. from individuals from n = 17 litters. Statistical analyses were performed by a two-tailed, unpaired, nonparametric Mann Whitney t test. * p ≤ 0.05; ** p ≤ 0.01. (**D**) Representative photography of four males littermates from a *Zdbf2*-GOF x *Zdbf2*-LOF cross (as shown in A) at 2 weeks of age. For each animal, genotype, dose of *Zdbf2* expression and weight are indicated.

*Zdbf2*-GOF/*Zdbf2*-LOF animals. However, we found that compared to single *Zdbf2*-LOF mutants, the presence of the maternal *Zdbf2*-GOF allele restored DNA methylation levels at the sDMR locus in all tissues of *Zdbf2*-GOF/*Zdbf2*-LOF mutants, with an average of 43.5% CpG methylation compared to the expected 50% in WT (*Duffié et al., 2014*; *Figure 3—figure supplement 2H*). By RT-qPCR, *Zdbf2* mRNA levels were also increased in the hypothalamus and pituitary gland of

*Zdbf2*-GOF/*Zdbf2*-LOF animals compared to single *Zdbf2*-LOF animals (*Figure 4B*), which strongly suggests that expression comes from the maternal *Zdbf2*-GOF allele. Accordingly, *Zdbf2* expression level in *Zdbf2*-GOF/*Zdbf2*-LOF animals was on average 0.65-fold the one of WT animals, which is congruent with the partial paternalization of the maternal *Zdbf2*-GOF allele we reported (*Figure 4B* and *Figure 3—figure supplement 2H*). Most importantly, restoration of *Zdbf2* expression by the maternal *Zdbf2*-GOF allele–even though incomplete–was sufficient to rescue the postnatal body weight phenotype in compound *Zdbf2*-GOF/*Zdbf2*-LOF males compared to their single *Zdbf2*-LOF brothers (*Figure 4C and D*). From a body weight reduction of 20% reported in *Liz*-LOF or *Zdbf2*-KO animals at the same age, it was attenuated to only 4% in *Zdbf2*-GOF/*Zdbf2*-LOF animals (*Figure 4C*), showing that maternal *Zdbf2* expression is as functional as paternal *Zdbf2* expression. Altogether, these results crystallize the importance of *Zdbf2* dosage in regulating postnatal body weight and most importantly, demonstrate that the dose but not the parental origin matters for *Zdbf2* function.

## *Zdbf2*-KO growth phenotype is correlated with decreased IGF-1 in the context of normal development of the hypothalamo-pituitary axis

Having demonstrated the growth promoting effect of *Zdbf2*, we next tackle the question of how it influences newborns weight and survival. As *Zdbf2* is expressed in the neuroendocrine cells of the hypothalamo-pituitary axis, the phenotype of *Zdbf2*-KO animals may lay in a defect in producing growth-stimulating pituitary hormones. When we assessed the development and functionality of the hypothalamus and pituitary gland prior to birth, we did not detect any morphological nor histological defects in *Zdbf2*-KO embryos (*Figure 5—figure supplement 1A-C*). Normal expression of major transcriptional regulators confirmed proper cell lineage differentiation in the *Zdbf2*-KO developing pituitary (*Figure 5—figure supplement 1B*; *Raetzman et al., 2002*; *Rizzoti, 2015*; *Kelberman et al., 2009*). Immunohistochemistry at E18.5 further indicated that *Zdbf2*-KO pituitary cells acquire normal competency for producing hormones (*Figure 5—figure supplement 1D*). Similarly, hypothalamic peptides were expressed in comparable levels in *Zdbf2*-KO and WT embryos, as assessed by in situ hybridization analysis (*Figure 5—figure supplement 1E-F*; *Biran et al., 2015*). In sum, the embryonic hypothalamo-pituitary axis develops normally in the absence of *Zdbf2*. Our data implies that the postnatal growth phenotype does not result from impaired establishment or programming of this axis during embryogenesis.

We then went on to analyze the functionality of the hypothalamo-pituitary axis in producing hormones after birth. Again, GH, ACTH, TSH, LH and PRL all appeared to be normally expressed in the pituitary glands of *Zdbf2*-KO juvenile animals (immunohistochemistry at 15dpp) (*Figure 5A*). Although we cannot exclude subtle dysfunctionalities, our results suggest that *Zdbf2* deficiency does not drastically compromise pituitary hormone production. Measured plasma GH levels in *Zdbf2*-KO animals also showed functional hormone release at 5 and 15dpp (*Figure 5B*).

However, *Zdbf2*-KO pups showed reduced plasma circulating levels of insulin growth factor 1 (IGF-1), a main regulator of postnatal growth, which reached only 70%, 45%, and 30% of WT level at 1dpp, 5dpp and 15dpp, respectively (*Figure 5C*). In adult mice, the liver is the main site of production of IGF-1, following transcriptional activation under the control of circulating GH (*Savage, 2013*). In contrast, extrahepatic production of IGF-1 during embryonic and early postnatal life is mostly GH-insensitive (*Lupu et al., 2001*; *Kaplan and Cohen, 2007*). As mentioned above, decreased IGF-1 levels in juvenile *Zdbf2*-KO animals seemed to occur in the context of normal GH input. Moreover, we measured normal *Igf1* mRNA levels by RT-qPCR in the liver of juvenile *Zdbf2*-KO animals, further illustrating that decreased IGF-1 levels are not a result of altered GH pathway (*Figure 5D*).

The association of low levels of circulating IGF-1 with normal GH secretion prompted us to evaluate more thoroughly the *Zdbf2*-KO growth phenotype. Mouse models of GH deficiency show growth retardation only from 10dpp onwards, while deficiency in IGF-1 affects growth earlier during postnatal development (*Lupu et al., 2001*). When we calculated the growth rate from day 1 to 8 weeks of age, we revealed two distinct phases: (*1*) from 1 to 7dpp, the *Zdbf2* mutant growth rate was 22% lower compared to WT littermates and (*2*) from 15 to 35dpp, the mutant exceeded the WT growth rate (*Figure 5E*) while no differences were observed in the *Zdbf2*-Δexon six silent mutation (*Figure 5—figure supplement 2A*). This shows that the growth defect is restricted to the first days of life. More specifically, at the day of birth (1dpp), *Zdbf2*-KO pups were smaller than their WT littermates by 9% (WT 1.41 ± 0.015 g, n = 99; *Zdbf2*-KO 1.29 ± 0.013, n = 95). After 1 week of postnatal life, the mutant

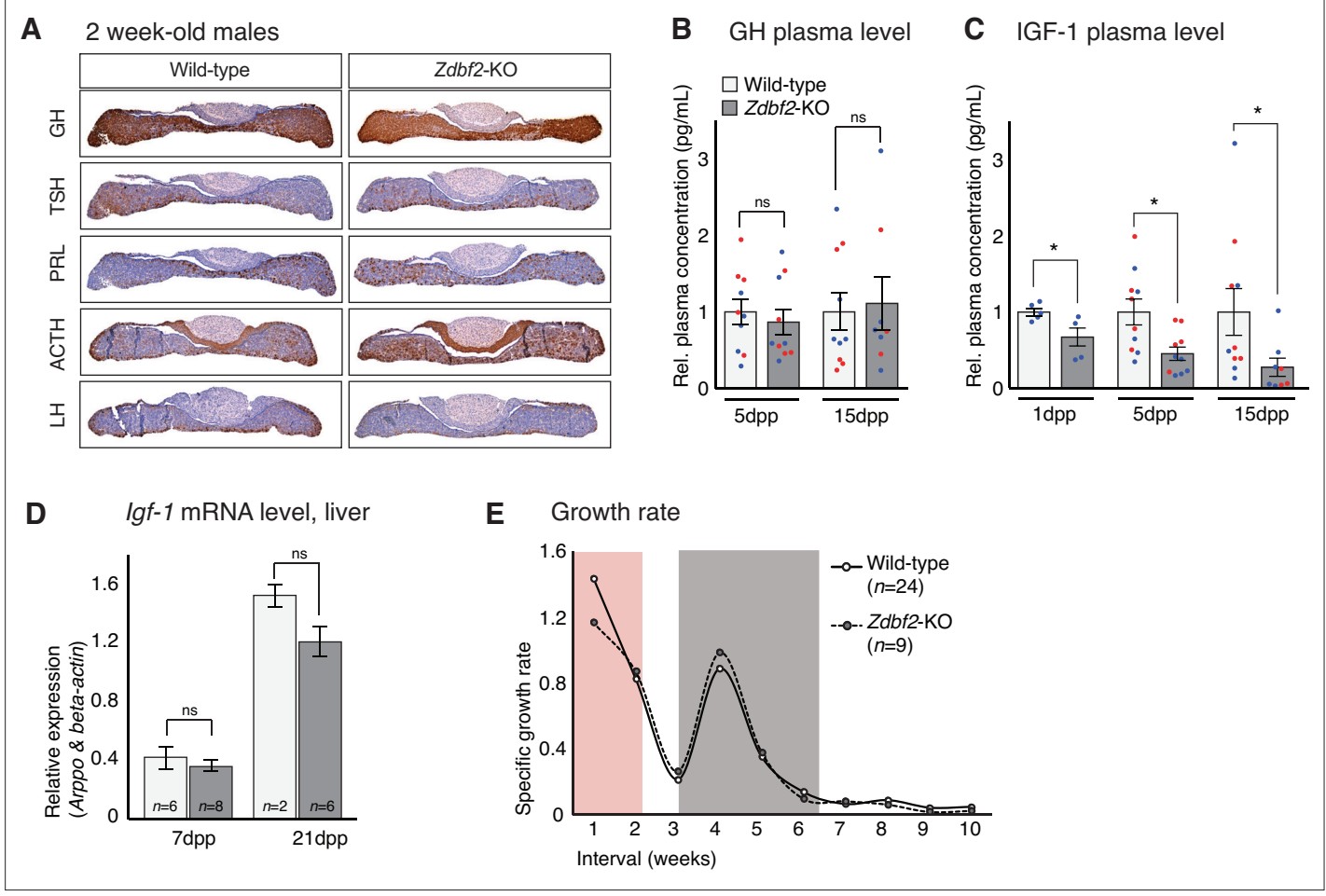

**Figure 5.** *Zdbf2*-KO phenotype is linked to defective IGF-1 signaling immediately after birth. (**A**) Pituitary hormone production is globally normal in *Zdbf2*-KO mice, as assessed by immunohistochemistry on 15dpp pituitary sections. (**B, C**) Circulating levels of GH at 5 and 15dpp (**B**) and of IGF-1 at 1, 5, and 15 dpp (**C**) in the plasma of WT and *Zdbf2*-KO mice. Data are shown as means ± s.e.m. of the relative expression to WT values from *n* replicates. Red and blue dots: females and males data points, respectively. Statistical analyses were performed by a two-tailed, unpaired, nonparametric Mann Whitney t test. *p ≤ 0.05. (**D**) RT-qPCR from postnatal liver measuring the level of *Igf-1* mRNAs between WT and *Zdbf2*-KO mice at 7 and 21dpp. Data are shown as means ± s.e.m from *n* WT and *Zdbf2*-KO animals. (**E**) Specific growth rate calculated from body weight of WT and *Zdbf2*-KO males from 1 to 10 weeks of age using the following equation: [(weight t2 - weight t1)/ weight t1]. Pink area: reduced growth rate in *Zdbf2*-KO pups compared to WT littermates; Grey area: growth rate of *Zdbf2*-KO mice exceeds the WT growth rate.

The online version of this article includes the following figure supplement(s) for figure 5:

**Figure supplement 1.** Normal development of the hypothalamo-pituitary axis in *Zdbf2*-KO mice.

**Figure supplement 2.** Specific growth rate in *Zdbf2*-Δexon6 and circulating IGF-1 in *Zdbf2*-GOF.

growth restriction reached 18% (WT 3.4 ± 0.13 g, n = 37; *Zdbf2*-KO 2.8 ± 0.11, n = 17), indicating a deficit in the ability to gain weight prior to 10dpp. Then, at 6 weeks of age, *Zdbf2*-KO animals were smaller than WT by only 8%, as a result of enhanced post-pubertal growth spurt (WT 21.7 ± 0.04 g, n = 34; *Zdbf2*-KO 20.05 ± 0.3, n = 15). We therefore concluded that the *Zdbf2*-KO phenotype is similar to defective IGF-1 signaling immediately after birth. However, circulating IGF-1 levels were not conversely increased in *Zdbf2*-GOF males at 5 and 15dpp (***Figure 5—figure supplement 2B***): IGF-1-independent mechanisms may be responsible for the larger body mass, or IGF-1 may be only transiently upregulated and not detected at these two timepoints.

Overall, we revealed that the body weight restriction of *Zdbf2*-KO juveniles is associated with a GH-independent decrease of IGF-1 during the first days of postnatal life, in the context of an overall normal development and functionality of the hypothalamo-pituitary axis.

## *Zdbf2*-KO neonates are undernourished and do not properly activate hypothalamic feeding circuits

Having determined that *Zdbf2*-KO animals are deficient in IGF-1, we attempted to define the molecular events associated with reduced IGF-1 levels. Undernutrition is a well-known cause of IGF-1 level reduction, and also of postnatal lethality (*Thissen et al., 1994*), which we observed in *Zdbf2*-KO pups. To investigate the nutritional status of *Zdbf2*-KO neonates, we weighed stomachs at 3dpp, as a measure of milk intake. *Zdbf2*-KO pups exhibited a significant reduction in stomach weight relative to body mass as compared to their WT littermates (*Figure 6A*), suggesting that these pups suffer milk deprivation. Stomach mass of *Zdbf2*-GOF males was not affected at 3dpp (*Figure 6—figure supplement 1A*), as expected from the normal body mass at the same age (*Figure 3D*). *Zdbf2*-KO pups showed normal relative mass at 3dpp for a range of organs at the exception of the interscapular brown adipose tissue (BAT), which was also reduced in *Zdbf2*-KO neonates (*Figure 6B* and *Figure 6—figure supplement 1B*). BAT-mediated thermogenesis regulates body heat during the first days after birth (*Cannon and Nedergaard, 2004*) and improper BAT function can lead to early postnatal death (*Charalambous et al., 2012*). However, despite being smaller, the BAT of *Zdbf2*-KO neonates appeared otherwise functional, showing normal lipid droplet enrichment on histological sections (data not shown) and proper expression of major markers of BAT thermogenic ability (*Figure 6—figure supplement 1C*). The BAT size reduction may therefore not reflect altered BAT ontogeny per se, but rather the nutritional deprivation of *Zdbf2*-KO neonates.

Because the mothers of *Zdbf2*-KO pups are of WT background, the undernutrition phenotype is unlikely due to defective maternal milk supply. Restoration of viability when placed in presence of KO-only littermates (*Figure 2K*) further indicated that *Zdbf2*-KO pups are competent to feed. Defective nutrition may therefore rather result from altered feeding motivation of the *Zdbf2*-KO neonates, specifically during the early nursing period. To test this hypothesis, we performed RNA-seq analysis of dissected hypothalami at 3dpp, when the growth phenotype is the most acute. Only 11 genes were significantly misexpressed in *Zdbf2*-KO hypothalamus relative to WT littermates (FDR 10%) (*Figure 6C* and *Figure 6—source data 1*). Only one of these genes is related to feeding, *Npy*, encoding the Neuropeptide Y, appearing as down-regulated in *Zdbf2*-KO hypothalamus. In line with our hypothesis, NPY is secreted from neurons of the hypothalamic arcuate nucleus and stimulates food intake and promotes gain weight (*Mercer et al., 2011*). Using our *Zdbf2*-LacZ reporter line, we revealed co-localization of beta-galactosidase and NPY staining in the hypothalamus (*Figure 6D*). This observation reinforces the possible involvement of ZDBF2 in NPY production in the paraventricular nucleus, to control food intake.

Little is known about the determinants of food intake in neonates (*Muscatelli and Bouret, 2018*); nonetheless, this prompted us to examine the expression levels of other hypothalamic modulators known to influence feeding behavior in adults. Interestingly, genes that positively regulate food intake tended to be down-regulated, while negative regulators of food intake were up-regulated (*Figure 6E*). One of these genes is the Agouti-related peptide (*Agrp*)-encoding gene (*Figure 6E*) that is co-expressed with *Npy* in a sub-population of hypothalamic neurons and convergently stimulate appetite and food seeking (*Gropp et al., 2005*). RT-qPCR measurement confirmed lower levels of both *Npy* and *Agrp* in *Zdbf2*-KO hypothalamus compared to WT, at 3dpp but also earlier on, at 1dpp (*Figure 6F*). It is noteworthy that none of the misregulated genes, including positive regulators of feeding, that we observed at 3dpp were also misregulated in the hypothalamus at 10dpp, as measured by RNA-seq (*Figure 6—figure supplement 1D-E*).

Together, these results provide key insights into the origin of the *Zdbf2* mutant phenotype: in absence of *Zdbf2*, the hypothalamic circuit of genes that stimulates food intake may not be properly activated at birth. This is associated with reduced milk intake, reduced body weight gain and suboptimal viability of *Zdbf2*-KO neonates when placed in presence of healthier littermates.

## Discussion

The hypothalamus is increasingly regarded as a major site for the action of imprinted genes on postnatal growth, feeding behavior and metabolism (*Ivanova and Kelsey, 2011*). Here, we found evidence that the imprinted *Zdbf2* gene stimulates hypothalamic feeding circuits, right after birth. In its absence, neonates do not ingest enough milk, and suffer from undernutrition. This leads to

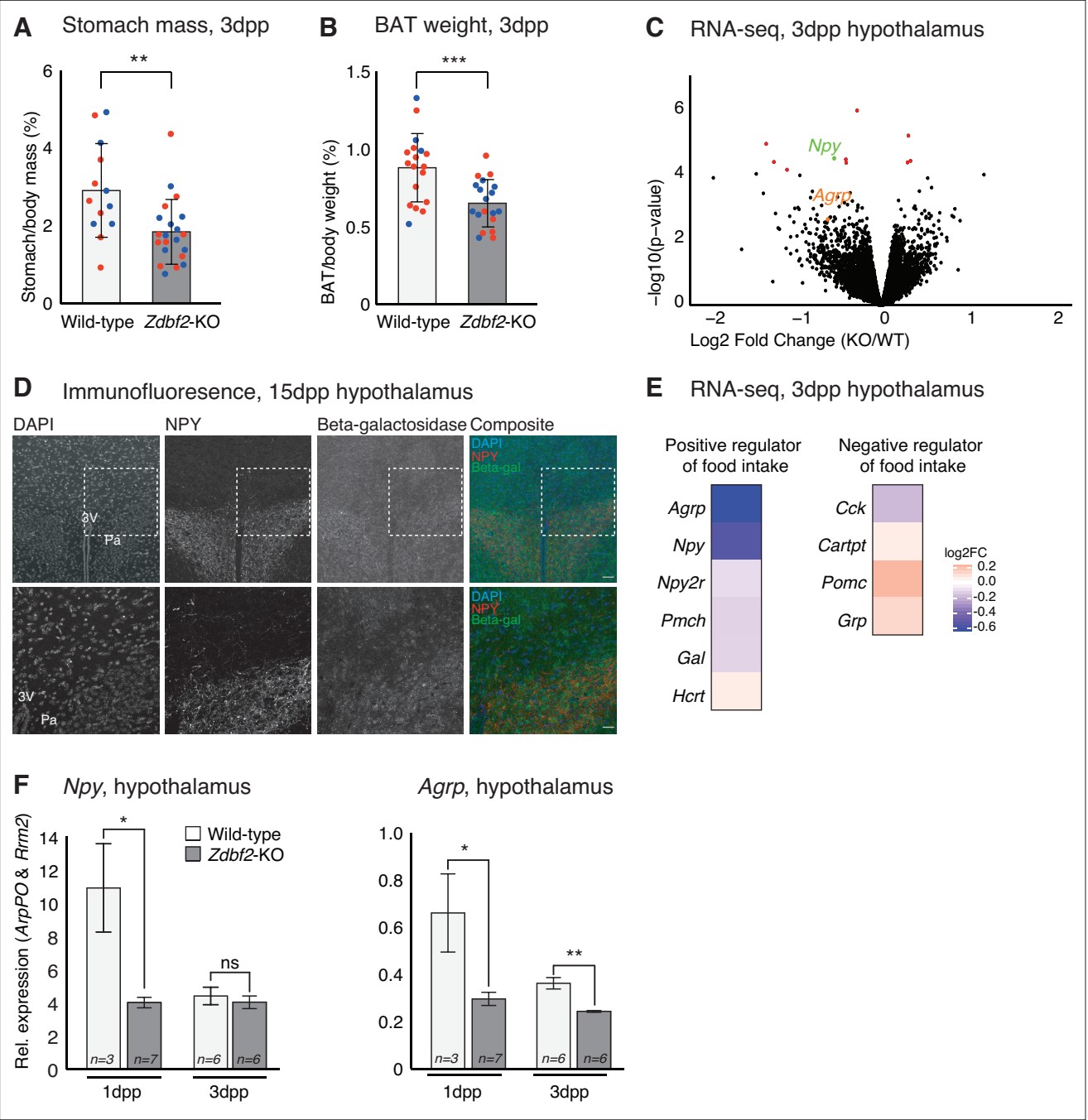

**Figure 6.** *Zdbf2*-KO neonates display a feeding defect. (**A, B**) Stomach (**A**) and brown adipocyte tissue (BAT) (**B**) mass normalized to the body mass for WT and *Zdbf2*-KO at 3dpp. Red dots: females; blue dots: males. Data are shown as means ± s.e.m. from *n* replicates. Statistical analyses were performed by a two-tailed, unpaired, nonparametric Mann Whitney t test.** p ≤ 0.01, ***p ≤ 0.005. (**C**) Volcano plot representation of RNA-seq of 3dpp hypothalamus of *Zdbf2*-KO versus WT littermates. n = 3 replicates for each genotype. Red dots: differentially expressed genes with a threshold of FDR < 10%. *Npy* (FDR 6%) is highlighted in green and *Agrp* in orange. (**D**) Representative image of immunofluorescence from brain sections, focused on hypothalamic region in *Zdbf2*:LacZ animals at 15dpp. Black and white images are shown for DAPI, NPY and Beta-galactosidase and composite images depict them in blue, red and green, respectively. Dotted square (top panel) represent the focused region in the bottom panel. Scale bar: 100 µm. *3* V, third ventricle; *Pa*, paraventricular hypothalamic nucleus. (**E**) Heatmap showing the log2 fold change of genes encoding hypothalamic regulators of food intake (RNA-seq data from C). (**F**) RT-qPCR from hypothalamus of 1 and 3dpp males animals measuring *Npy* (left panel) and *Agrp* (right panel) mRNA levels. Data are shown as means ± s.e.m. from *n* replicates. Statistical analyses were performed by a two-tailed, unpaired, nonparametric Mann Whitney t test. *p ≤ 0.05, **p ≤ 0.01.

The online version of this article includes the following source data and figure supplement(s) for figure 6:

*Figure 6 continued on next page*

*Figure 6 continued*

**Source data 1.** List of differentially expressed genes in the hypothalamus of *Zdbf2*-KO versus WT males at 3dpp and 10dpp.

**Figure supplement 1.** Characterization of the early postnatal feeding behavior in *Zdbf2*-KO pups.

decreased IGF-1 signaling within the first week of life, reduced body weight gain and more dramatically, lethality of half of the *Zdbf2*-KO pups when they are in presence of healthier WT littermates. Our work therefore highlights that *Zdbf2* is necessary to thrive and survive after birth, by allowing newborns to adapt to postnatal feeding. Interestingly, *Zdbf2* expression is persistently high in adult brains, which may point to other functions in later life.

By relying on a unique collection of mouse models of total loss of function (*Zdbf2*-KO and *Liz*-LOF), partial loss of function (*Zdbf2*-lacZ), normal function (*Zdbf2*-WT), and gain of function (*Zdbf2*-GOF), we observed that postnatal body growth is exquisitely sensitive to the quantity of *Zdbf2* produced in the hypothalamo-pituitary axis. Incidentally, *Zdbf2* meets the criteria of a *bona fide* growth-promoting gene: growth is reduced upon decreased dosage and oppositely enhanced upon increased dosage of *Zdbf2* (*Efstratiadis, 1998*). Consistent with previous results for the imprinted *Cdkn1c* gene (*Andrews et al., 2007*), growth reduction was more pronounced than overgrowth upon changes in *Zdbf2* expression (18% decrease and 10% increase at 10dpp) and specific to males. This reflects that, with standard diet, overgrowth is a much less frequent response than growth reduction and affects animals with favorable physiology only, potentially explaining male specificity.

With a few exceptions, most studies have only addressed the phenotypic effects of reducing the dose of an imprinted gene, but not what results from overexpressing this same gene (*Tucci et al., 2019*). Knocking out an imprinted gene provides valuable insights about its physiological function; however, it does not address the evolutionary significance of imprinting of that gene, that is reduction to mono-allelic expression. With the *Zdbf2*-GOF model, we were able to evaluate the consequences of a loss of *Zdbf2* imprinting: bi-allelic and increased *Zdbf2* expression exacerbates postnatal body weight gain. However, despite being slightly bigger, viability, fertility or longevity appeared overall normal in *Zdbf2*-GOF animals. The evolutionary importance of *Zdbf2* imprinting is therefore not immediately obvious, at least under unchallenged conditions. Finally, even fewer studies have addressed the importance of parent-of-origin expression for imprinted gene functions (*Drake et al., 2009*; *Leighton et al., 1995*). By intercrossing models of *Zdbf2* loss and gain of function, we could enforce *Zdbf2* expression from the maternal allele while inactivating the normally expressed paternal allele. Restoration of body weight demonstrated the functionality of maternal *Zdbf2* expression, reinforcing that *Zdbf2* functions on postnatal body homeostasis in a dose-dependent manner. Although not necessarily surprising, this demonstration is conceptually important for understanding the evolution and *raison d'être* of genomic imprinting. The divergent DNA methylation patterns that are established in the oocyte and the spermatozoon provide opportunities to evolve mono-allelic regulation of expression, but once transmitted to the offspring, the parental origin of expression is not essential per se.

Although *Zdbf2* is expressed across the hypothalamo-pituitary axis, we could not find evidence of abnormal development or function of the pituitary gland that could explain the *Zdbf2*-KO growth phenotype. Notably, GH production and release were normal, at least from 5dpp and on. Additionally, the *Zdbf2* growth reduction diverges from GH-related dwarfism: *Gh*-deficient mice grow normally until 10dpp, after which only they exhibit general growth impairment with reduced levels of circulating IGF-1 (*Voss and Rosenfeld, 1992*; *Lupu et al., 2001*). In *Zdbf2*-KO mutants, the growth defect is apparent as soon as 1dpp, as well as decreased IGF-1. In *Igf-1* null mice, birthweight is approximately 60% of normal weight and some mutants die within the first hours after birth (*Liu et al., 1993*; *Efstratiadis, 1998*). The *Zdbf2*-KO phenotype thus resembles an attenuated *Igf-1* deficiency, in agreement with half reduction but not total lack of circulating IGF-1. Interestingly, similar IGF-1-related growth defects have been reported upon alteration of other imprinted loci in the mouse: the *Rasgrf1* gene, the *Dlk1-Dio3* cluster and the *Cdkn1c* gene (*Itier et al., 1998*; *Andrews et al., 2007*; *Charalambous et al., 2014*). However, the origin of IGF-1 deficiency may be different: in *Rasgrf1* mutants, unlike *Zdbf2* mutants, this was linked to impaired hypothalamo-pituitary axis and GH misregulation (*Drake et al., 2009*). Finally, decreased *ZDBF2* levels have recently been associated with intra-uterine growth restriction (IUGR) in humans (*Monteagudo-Sánchez et al., 2019*). Whether this is linked to impaired

fetal IGF-1 production or to distinct roles of ZDBF2 related to placental development in humans would be interesting to assess, in regards to conservation or not of imprinted gene function across mammals.

IGF-1 secretion has been shown to drop in response to starvation, leading to disturbed growth physiology (*Savage, 2013*). Our findings support that limited food intake is probably the primary defect in *Zdbf2* deficiency, leading to IGF-I insufficiency in the critical period of postnatal development and consequently, growth restriction. First, we showed that *Zdbf2* is expressed in hypothalamic regions that contain neurons with functions in appetite and food intake regulation, such as the arcuate and paraventricular nucleus. Second, *Zdbf2-KO* neonates show hypothalamic downregulation of the *Npy* and *Agrp* genes that encode for orexigenic neuropeptides (*Stanley and Leibowitz, 1984*; *Ollmann et al., 1997*). These are likely direct effects: (i) NPY and *Zdbf2*-LacZ are co-expressed in the same cells and (ii) the rest of the hypothalamic transcriptome is scarcely modified in *Zdbf2*-KO pups. Quantified changes were not of large magnitude, but AgRP/NPY neurons represent a very small population of cells, present in the arcuate nucleus of the hypothalamus only (*Andermann and Lowell, 2017*). We are likely at the limit of detection when analyzing these genes in the whole hypothalamus transcriptome. Finally, *Npy* and *Agrp* downregulation was observed immediately at birth, along with a phenotype of reduced milk consumption, as measured by stomach weighing. Given our observations, we propose that ZDBF2 activates specialized hypothalamic neurons that motivate neonates to actively demand food (milk) from the mother right at birth, promoting the transition to oral feeding after a period of passive food supply in utero. This function agrees with the co-adaptation theory according to which genomic imprinting evolved to coordinate interactions between the offspring and the mother (*Wolf and Hager, 2006*). Our results are also in line with the kindship theory of genomic imprinting (*Haig, 2000*): *Zdbf2* is a paternally expressed gene that potentiates resource extraction from the mother. Finally, despite considerable effort, we were unable to specifically detect the ZDBF2 protein with antibodies or using epitope-tagging approaches of the endogenous gene; future studies hopefully will bring clarity to which molecular function ZDBF2 carries in the mouse hypothalamus.

In conclusion, we reveal here that decreasing *Zdbf2* compromises resource acquisition and body weight gain right after birth. Restricted postnatal growth can have a strong causal effect on metabolic phenotypes, increasing the risk of developing obesity in later life. This is observed in mouse KO models of the imprinted *Magel2* gene that map to the Prader Willi syndrome (PWS) region, recapitulating some features of PWS patients, who after a failure to thrive as young infants exhibit a catch-up phase leading to overweight and hyperphagia (*Bischof et al., 2007*). *Zdbf2*-KO mice do present a post-puberal spurt of growth, which attenuates their smaller body phenotype, but we never observed excessive weight gain, even after 18 months (data not shown). It would be interesting to test whether the restricted postnatal growth of *Zdbf2*-KO mice may nonetheless increase the likelihood of metabolic complications when challenged with high-fat or high-sugar diet.

## Materials and methods

**Key resources table**

| Reagent type (species) or resource | Designation | Source or reference | Identifiers | Additional information |
|---|---|---|---|---|
| Genetic reagent (Mus. Musculus) | *Zdbf2*-KO | This study | | CRISPR/Cas9 generated mutant, sgRNA oligos are listed in *Supplementary file 1* |
| Genetic reagent (Mus. Musculus) | *Zdbf2*-GOF | Bourc'his lab | | *Greenberg et al., 2017* |
| Genetic reagent (Mus. Musculus) | *Zdbf2*-LacZ reporter line | Bourc'his lab | EUCOMM Project Number: *Zdbf2_82543* | *Greenberg et al., 2017* |
| Genetic reagent (Mus. Musculus) | *Liz*-LOF | Bourc'his lab | | *Greenberg et al., 2017* |
| Antibody | Anti-ACTH, mouse monoclonal | Fitzgerald | RRID:AB_1282437 | Ref. 10C-CR1096M1, 1:1,000 |
| Antibody | Anti-GH, rabbit polyclonal | National Hormone and Peptide Program (*NHPP*) | | Ref. AFP-5641801, 1:1,000 |

*Continued on next page*

*Continued*

| Reagent type (species) or resource | Designation | Source or reference | Identifiers | Additional information |
|---|---|---|---|---|
| Antibody | Anti-TSH, rabbit polyclonal | National Hormone and Peptide Program (*NHPP*) | | Ref. AFP-1274789, 1:1,000 |
| Antibody | Anti-PRL, rabbit polyclonal | National Hormone and Peptide Program (*NHPP*) | | Ref. AFP-425-10-91, 1:1,000 |
| Antibody | Anti-LH, rabbit polyclonal | National Hormone and Peptide Program (*NHPP*) | | Ref. AFP-C697071P, 1:500 |
| Antibody | Anti-NPY, rabbit polyclonal | Cell Signaling Technology | RRID:AB_2716286 | Ref. # 11976, 1:1,000 |
| Antibody | Anti-beta-galactosidase, chicken polyclonal | Abcam | | Ref. # ab9361, 1:1,000 |
| Antibody | Goat anti-rabbit Alexa-fluorophore 594 | Invitrogen | RRID:AB_2762824 | Ref. #A32740, 1:1,000 |
| Antibody | Gao anti-chicken Alexa-fluorophore 488 | Invitrogen | RRID:AB_2534096 | Ref. # A11039, 1:1,000 |
| Commercial assay or kit | Mouse Magnetic Luminex Assay for IGF-1 | R&D System | | |
| Commercial assay or kit | Milliplex Mouse Pituitary Magnetic Assay for GH | Merck | | |
| Software, algorithm | STAR_2.6.1 a | *Dobin et al., 2013* | | |

## Mice

Mice were hosted on a 12 hr/12 hr light/dark cycle with free access to food and water in the pathogen-free Animal Care Facility of the Institut Curie (agreement number: C 75-05-18). All experimentation was approved by the Institut Curie Animal Care and Use Committee and adhered to European and National Regulation for the Protection of Vertebrate Animals used for Experimental and other Scientific Purposes (Directive 86/609 and 2010/63). For tissue and embryo collection, euthanasia was performed by cervical dislocation. The *Zdbf2*-KO and *Zdbf2*-GOF mutant mice lines were derived by CRISPR/Cas9 engineering in one-cell stage embryos as previously described (*Greenberg et al., 2017*), using two deletion-promoting sgRNAs. Zygote injection of the CRISPR/Cas9 system was performed by the Transgenesis Platform of the Institut Curie. Eight week-old superovulated C57BL/6 J females were mated to stud males of the same background. Cytoplasmic injection of Cas9 mRNA and sgRNAs (100 and 50 ng/µl, respectively) was performed in zygotes collected in M2 medium (Sigma) at E0.5, with well-recognized pronuclei. Injected embryos were cultured in M16 medium (Sigma) at 37 °C under 5% CO2, until transfer at the one-cell stage the same day or at the two-cell stage the following day in the infudibulum of the oviduct of pseudogestant CD1 females at E0.5. The founder mice were then genotyped and two independent founders with the expected deletion were backcrossed to segregate out undesired genetic events, with a systematic breeding scheme of *Zdbf2*-KO heterozygous females x WT C57Bl6/J males and *Zdbf2*-GOF heterozygous males x WT C57Bl6/J females to promote silent passing of the deletion. Cohorts of female and male N3 animals were then mated with WT C57Bl6/J to study the maternal and paternal transmission of the mutation.

The LacZ-*Zdbf2* reporter line was derived from mouse embryonic stem (ES) cells from the European Conditional Mouse Mutagenesis Program (EUCOMM Project Number: *Zdbf2*_82543). Proper insertion of the LacZ construct was confirmed by long-range PCR. However, we found that the loxP site in the middle position was mutated (A to G transition at position 16 of the loxP site) in the original ES cells (*Figure 3—figure supplement 1A*). Chimeric mice were generated through blastocyst injection by the Institut Curie Transgenesis platform. We studied animals with an intact LacZ-KI allele, without FRT- or CRE-induced deletions.

## DNA methylation analyses

Genomic DNA from adult tissues was obtained following overnight lysis at 50 °C (100 mM Tris pH 8, 5 mM EDTA, 200 mM NaCl, 0.2% SDS and Proteinase K). DNA was recovered by a standard phenol/choloroform/isoamyl alcohol extraction and resuspended in water. Bisulfite conversion was performed

on 0.5–1 μg of DNA using the EpiTect Bisulfite Kit (Qiagen). Bisulfite-treated DNA was PCR amplified and either cloned and sequenced, or analyzed by pyrosequencing. For the former, 20–30 clones were Sanger sequenced and analyzed with BiQ Analyzer software (*Bock et al., 2005*). Pyrosequencing was performed on the PyroMark Q24 (Qiagen) according to the manufacturer's instructions, and results were analyzed with the associated software.

## RNA expression analyses

Total RNA was extracted using Trizol (Life Technologies). To generate cDNA, 1 μg of Trizol- extracted total RNA was DNase-treated (Ambion), then reverse transcribed with SuperscriptIII (Life Technologies) primed with random hexamers. RT-qPCR was performed using the SYBR Green Master Mix on the ViiA7 Real-Time PCR System (Thermo Fisher Scientific). Relative expression levels were normalized to the geometric mean of the Ct for housekeeping genes *Rrm2*, *B-actin* and/or *Rplp0*, with the ΔΔCt method. Primers used are listed in *Supplementary file 1*.

For RNA-sequencing, hypothalami of three animals at 3dpp and two animals at 10dpp were collected for each genotype (all males) and RNA was extracted. Trizol-extracted total RNA was DNase-treated with the Qiagen RNase-Free DNase set, quantified using Qubit Fluorometric Quantitation (Thermo Fisher Scientific) and checked for integrity using Tapestation (Agilent). RNA-seq libraries were cloned using TruSeq Stranded mRNA LT Sample Kit on total RNA (Illumina) and sequencing was performed on a NovaSeq 6,000 (Illumina) at the NGS platform of the Institut Curie (PE100, approximately 35M to 50M clusters per replicate for 3dpp, and 16–30 M for 10dpp).

## LacZ staining

Whole brains and pituitary glands were fixed in 4% paraformaldehyde (PFA) in PBS (pH 7.2) overnight and washed in PBS. For sections, tissues were then incubated in sucrose gradients and embedded in OCT for conservation at –80 °C before cryosectioning. Sections were first fixed 10 min in solution of Glutaraldehyde (0.02% Glutaraldehyde, 2 mM MgCl2 in PBS). Tissues and sections were washed in washing solution (2 mM MgCl2, 0.02% NP40, 0.01% C24H39NaO4 in PBS) and finally incubated at room temperature overnight in X- gal solution (5 mM K4Fe(CN)63H20, 5 mM K3Fe(CN)6, 25 mg/mL X-gal in wash solution). After several PBS washes, sections were mounted with an aqueous media before imaging. N = 3 biological replicates were tested.

## Histological analysis and RNA in situ hybridization

Embryos at E13.5 and E15.5 were embedded in paraffin and sectioned at a thickness of 5 μm. For histological analysis, paraffin sections were stained with Hematoxylin and Eosin. RNA in situ hybridization analysis on paraffin sections was performed following a standard procedure with digoxigenin-labeled antisense riboprobes as previously described (*Gaston-Massuet et al., 2008*).The antisense riboprobes used in this study [α-Gsu, *Pomc1*, *Lhx3*, *Pitx1*, *Avp*, oxytocin and *Ghrh*] have been previously described (*Gaston-Massuet et al., 2008*; *Gaston-Massuet et al., 2016*). N = 3 biological replicates were tested.

## Immunohistochemistry on histological sections

Embryos were fixed in 4% PFA and processed for immuno-detection as previously described (*Andoniadou et al., 2013*). Hormones were detected using antibodies for α-ACTH (mouse monoclonal, 10C-CR1096M1, RRID:AB_1282437, 1:1000), α-GH (rabbit polyclonal, NHPP AFP-5641801, 1:1000), α-TSH (rabbit polyclonal, NHPP AFP-1274789, 1:1000), α-PRL (rabbit polyclonal, NHPP AFP-425-10-91, 1:1000), and α-LH (rabbit polyclonal, NHPP AFP-C697071P, 1:500). N = 3 biological replicates were tested.

## Immunofluorescence on hypothalamus sections

Cryostat sections were washed with PBS, permeabilized and blocked with Blocking buffer (5% horse serum, 3% BSA, 0.2% Triton X-100 in PBS) prior to primary antibody incubation at 4 °C overnight. Secondary antibody staining was performed for one hour and DAPI for 5 min at room temperature. Finally, slides were mounted with Vectashield mounting media and imaged with a confocal microscope LSM700. The following pairs of primary and secondary antibodies were used: anti-NPY (rabbit polyclonal, Cell Signaling Technology Cat# 11976, RRID:AB_2716286, 1:1000) with goat anti-rabbit

Alexa-fluorophore 594 (1:1000; Invitrogen Cat# A32740, RRID: AB_2762824) and anti-β-galactosidase (chicken polyclonal, Abcam Cat# ab9361, RRID:AB_307210, 1:1000) with goat anti-chicken Alexa-fluorophore 488 (1:1000; Invitrogen Cat# A11039, RRID: AB_2534096). N = 3 biological replicates were tested.

### LUMINEX ELISA assay

Plasma from blood was collected in the morning at fixed time on EDTA from 1, 5 and 15 day-old mice after euthanasia and stored at –20°C until use. Samples were run in duplicates using the Mouse Magnetic Luminex Assay for IGF-1 (R&D System) and Milliplex Mouse Pituitary Magnetic Assay for GH (Merck) according to manufacturers' instructions. Values were read on Bio-Plex 200 (Bio-Rad) and analyzed with the Bio-Plex Manager Software.

### Phenotypic analyses of weight

Postnatal weight measurements were performed every two days from 1dpp to weaning age (21 days). Then, mice were separated according to their genotype, hosted in equal number per cage (n = 5–6) and weighted once per week. As body weight is a continuous variable, we used a formula derived from the formula for the $t$-test to compute the minimum sample size per genotype: n = 1 + C(s/d)2, where $C$ is dependent on values chosen for significance level ($\alpha$) and power (1-$\beta$), $s$ is the standard deviation and $d$ the expected difference in means. Using $\alpha$ = 5%, $\beta$ = 90% and an expected difference in means of 1 g at 2 weeks and 2 g later on, we predicted a minimum $n$ of between 10 and 20 and thus decided to increase this number using $n$ size between 20 and 30 per genotype, depending of the age and the sex of the mice. No animals were excluded from the analysis. Animals were blindly weighed until genotyping. E18.5 embryos and postnatal organs were collected and individually, rinsed in PBS and weighted on a 0.001 g scale. All data were generated using three independent mating pairs.

### Phenotypic analyses of postnatal lethality

Material for genotyping was taken the first day of birth and then number of pups for a given litter was assessed every day. To avoid bias, we excluded from the analysis the litters where all the pups died due to neglecting mothers, not taking care of their pups.

### DEXA scan analyses

The DEXA analysis allows the assessment of fat and lean mass, bone area, bone mineral content, and bone mineral density. Practically, mutant and WT littermate males at 2 weeks were sent from the Animal Facility of Institut Curie to the Mouse Clinics along with their mother (Ilkirch, France). The phenotypic DEXA analysis was performed at 7 weeks of age using an *Ultrafocus DXA* digital radiography system, after the mandatory 5-week-quarantine period in the new animal facility. Mice were anesthetized prior analysis and scarified directly after measurements, without a waking up phase.

### RNA- Seq data analysis

Adapters sequences were trimmed using TrimGalore v0.6.2 (https://github.com/FelixKrueger/TrimGalore; *Krueger, 2022*). N = 3 biological replicates were sequenced per genotype at 3dpp, and n = 2 at 10dpp. Paired-end reads were mapped using STAR_2.6.1 a (*Dobin et al., 2013*) allowing 4% of mismatches. Gencode vM13 annotation was used to quantify gene expression using quantification mode from STAR. Normalization and differential gene expression were performed using EdgeR R package (v3.22.3) (*Robinson et al., 2010*). Genes were called as differentially expression if the fold discovery rate (FDR) is lower than 10%.

### Statistical analyses

Significance of obtained data was determined by performing one-way or two-way ANOVA test or two-tailed unpaired, nonparametric Mann Whitney $t$- tests using GraphPad Prism6 software. p values were considered as significant when p ≤ 0.05. Data points are denoted by stars based on their significance: ****: p ≤ 0.0001; ***: p ≤ 0.001; **: p ≤ 0.01; *: p ≤ 0.05.

## Acknowledgements

We would like to thank members of the Bourc'his laboratory for continuous support and stimulation, M Greenberg for critical reading of the manuscript, M Charalambous for methodological and conceptual advices, T Chelmicki and L Marion-Poll for technical help and F El Marjou (Transgenesis Platform of the Institut Curie Animal Facility) for CRISPR in vivo engineering. Luminex assays were performed by the LUMINEX Platform from the CHU Clermont-Ferrand. The laboratory of DB is part of the Laboratory d'Excellence LABEX (LABEX) entitled DEEP (11-LBX0044). This research was supported by the ERC (grant ERC-Cog EpiRepro) and the Bettencourt Schueller Foundation. CGM and AG were supported by grants from Action Medical Research (GN2272) and BTL Charity (GN417/2238). JG was a recipient of PhD fellowships from DIM Biothérapies, Ile-de-France and from La Ligue contre le Cancer.

# Additional information

## Competing interests
Deborah Bourc'his: Reviewing editor, eLife. The other authors declare that no competing interests exist.

## Funding

| Funder | Grant reference number | Author |
|---|---|---|
| FP7 Ideas: European Research Council | ERC-Cog EpiRepro | Aurélie Teissandier<br>Deborah Bourc'his |
| Fondation Bettencourt Schueller | | Deborah Bourc'his |
| Ligue Contre le Cancer | | Juliane Glaser |
| BTL Charity | GN417/2238 | Angelica Gualtieri<br>Carles Gaston-Massuet |
| Action Medical Research | GN2272 | Angelica Gualtieri<br>Carles Gaston-Massuet |
| DIM Biotherapies | | Juliane Glaser |

The funders had no role in study design, data collection and interpretation, or the decision to submit the work for publication.

## Author contributions
Juliane Glaser, Conceptualization, Formal analysis, Investigation, Methodology, Resources, Validation, Visualization, Writing – original draft, Writing – review and editing; Julian Iranzo, Investigation, Validation; Maud Borensztein, Mattia Marinucci, Investigation; Angelica Gualtieri, Formal analysis, Investigation; Colin Jouhanneau, Methodology; Aurélie Teissandier, Formal analysis, Visualization; Carles Gaston-Massuet, Funding acquisition, Visualization, Writing – original draft; Deborah Bourc'his, Conceptualization, Formal analysis, Funding acquisition, Supervision, Writing – original draft, Writing – review and editing

## Author ORCIDs
Juliane Glaser http://orcid.org/0000-0001-6745-6924
Julian Iranzo http://orcid.org/0000-0002-9369-2530
Maud Borensztein http://orcid.org/0000-0002-4378-5018
Deborah Bourc'his http://orcid.org/0000-0001-9499-7291

## Ethics
All experimentation was approved by the Animal Care and Use Committee of the Institut Curie (agreement number: C 75-05-18) and adhered to European and National Regulation for the Protection of Vertebrate Animals used for Experimental and other Scientific Purposes (Directive 86/609 and 2010/63).

## Decision letter and Author response
Decision letter https://doi.org/10.7554/eLife.65641.sa1

Author response https://doi.org/10.7554/eLife.65641.sa2

## Additional files

### Supplementary files
- Transparent reporting form
- Supplementary file 1. List of primers used in this study.

### Data availability
All data generated or analysed during this study are included in the manuscript and supporting files. RNA-Seq data have been deposited in GEO under accession code GSE153265.

The following dataset was generated:

| Author(s) | Year | Dataset title | Dataset URL | Database and Identifier |
|---|---|---|---|---|
| Glaser J, Iranzo J, Borensztein M, Marinucci M, Gualtieri A, Jouhanneau C, Teissandier A, Gaston-Massuet C, Bourc'his D | 2021 | RNA-seq of whole hypothalamus in WT and Zdbf2-KO neonates | https://www.ncbi.nlm.nih.gov/geo/query/acc.cgi?acc=GSE153265 | NCBI Gene Expression Omnibus, GSE153265 |

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
