## [Editor Report]

The paper provides an elegant demonstration of the phenotypic consequences of genomic imprinting for postnatal physiology, focusing on the specific case of the Zdbf2 gene in the mouse. Using a series of gain and loss function models they explore how manipulating gene dosage independently of the parent of origin, influences the hypothalamic-pituitary endocrine axis, to regulate feeding behavior and growth during the early post-natal period.

---

## [Decision Letter]

**Decision letter after peer review:**

Thank you for submitting your article "The imprinted Zdbf2 gene finely tunes feeding and growth in neonates" for consideration by *eLife*. Your article has been reviewed by 3 peer reviewers, and the evaluation has been overseen by a Reviewing Editor and Carlos Isales as the Senior Editor. The following individuals involved in review of your submission have agreed to reveal their identity: Andrew Ward (Reviewer #1); Stormy Chamberlain (Reviewer #2); Gavin Kelsey (Reviewer #3).

Essential revisions:

1) Improved statistics and analysis of the growth phenotype. Reviewer 1 makes some excellent suggestions that could improve the manuscript.

2) An attempt to probe mechanism based on better measurement of IGF and GH levels, including Reviewer 2's point about IGF levels in pups with wild type and mutant littermates. The authors should provide a more careful time course of IGF-1 levels in the various genotypes (both gain and loss of function) and littermate competition situations. Reviewer 3 also points to how immunofluorescence studies could be used to characterize how the different mutants effect the population of AgRP/NPY neurons, to better correlate a cellular phenotypes with the growth ones.

3) A better understanding of the Zdbf2 molecular mechanism. RNAseq demonstrated down-regulation of transcripts for AgRP and NPY in early postnatal hypothalamus. Is this a direct effect on the regulation of AgRP and NPY transcription or impaired development or maturation of AgRP/NPY neurons? Immunofluorescence studies enable an estimate to be made of the AgRP/NPY neuron population to identify any deficiencies in Zbf2 mutants (see above).

4) Better analysis of Zbdf2 expression. Review 3 makes a good suggestion, using β galactosidase staining based on the gene trap allele.

5) Specific revisions to the discussion as suggested by the reviewers.

*Reviewer #1:*

In this paper both gain- and loss-of-function mouse genetic models are used to characterise a post-natal growth phenotype associated with altered dosage of the imprinted Zdbf2 gene, which is expressed in the hypothalamus and pituitary gland from the paternally-inherited allele. A growth deficit in the immediate post-natal period is identified in Zdbf2-KO animals and mirrored in animals with increased Zdbf2 expression. Some details of the growth phenotypes could be more accurately and fully described but overall there is good evidence that Zdbf2, through expression in specific hypothalamic nuclei and the pituitary gland, promotes suckling in pups during the immediate post-partum period. Intriguingly, the transient growth deficiency appears to affect adult lean body mass and brown adipose tissue (BAT), rather than white adipose (WAT) deposition, though this needs to be more firmly established. If correct, this effect is unlike most early-life programming effects, in which disruptions in growth tend to result in excess WAT deposition in later life.

A genetic experiment, combining both gain- and loss-of function alleles to essentially switch expression of Zdbf2 from the paternal to the maternal allele shows that dosage, and not the parental allele origin of expression per se, is what matters. Though interesting, the outcome is unsurprising, as the authors themselves remark, and I would urge caution in using it to draw conclusions about the evolutionary mechanisms that have driven the evolution of imprinting.

Further experiments link the early changes in body size with a deficiency in Igf1 that may be due to reduced nutrient intake very early post-partum. These observations, together with transcriptomic data indicting altered expression of two genes (among only 11 with significantly different expression levels) known to have roles in hypothalamic regulation of feeding behaviour, comprise a compelling description of the role of imprinted Zdbf2 as a regulator of the hypothalamic feeding circuit. As a paternally-expressed gene that promotes growth and nutrient acquisition from the mother Zdbf2 thereby fits the parental conflict hypothesis for imprinted gene evolution. It may also fit with the mother-offspring co-adaptation theory, as suggested in the Discussion, but evidence for this is incomplete. Continued expression of Zdbf2 in the adult brain begs questions about a potential role in controlling behaviour in later life and were it found to influence maternal nurturing behaviour or milk let-down, for instance, this would better support the idea of co-adaptive evolution.

The authors establish that mouse Zdbf2 expression is restricted to hypothalamus and pituitary using both data generated themselves and that from publically available transcriptomic datasets (Figures1 and S1). However, none of the analysis rules out expression of Zdbf2 elsewhere during fetal or early post-natal life and I wonder if this has been established elsewhere? It is important because expression in, for example, skeletal muscle could provide an alternative explanation for the described phenotype. Also, they show levels of expression in both hypothalamus and pituitary arising during the pre-weaning period, between PN4 and PN21, then remaining high well into adulthood (PN70) (Figure 1E,F). Given that the major effect of Zdbf2 KO on growth is later shown to occur immediately after birth it would strengthen the paper to show expression during this period (PN1-3) and earlier, in the fetal brain. In addition, they might comment in the Discussion on the persistent and relatively high level of expression seen in adults, which points to further functions of Zdbf2 in later life, perhaps relating to behaviour.

Zdbf2 KO pups are shown to have a deficit in their ability to gain weight in the immediate post-natal period, in comparison with WT littermates (Figure 2). My major concern here is the statistical treatment of the data: No test appears to have been applied to data in Figure 2B and for that in Figures2C and 2D the use of a Mann Whitney t-test, presumably applied repeatedly to pairs of samples at each time-point, is inappropriate. Instead, ANOVA or similar should be used. Also, the description of the data is inaccurate: "Growth reduction was the most drastic prior to weaning, from 5 to 21dpp" (Lines 176-177). In fact, KO pups gain less weight than their WT littermates during the period PN1-PN5. It would be better to describe the differences in mass between WT and KO animals that the graphs show, rather than growth which is inferred (and more properly analysed later on in Figure 5E). It is potentially confusing to talk about a "growth reduction" in KO pups because the mass of both WT and KO animals increases over time (i.e. both are growing but the KOs less than normal). It would be more accurate to say, e.g. "while there is essentially no difference in mass at e18.5, KOs appear smaller than WT littermates at PN1 and PN5, indicating slower growth in the first few days after birth. The lower mass of KO animals persists into adulthood, measured up to 12 weeks of age". Here, I am also taking issue with the statement that the body weight reduction had "a tendency to minimize with age (Figure 2B)" (Line 179) because any difference in Figure 2B is subtle and the body mass difference is still evident post weaning (Figure 2C) and is not obviously reduced.

I would be more cautious with the statement, "The reduced body weight phenotype was fully penetrant (Figure S2E)." First, the figure legend says 'highly' penetrant, which seems a better term to use because there are animals with mass equal to or greater than mean WT mass. Also, penetrance is difficult to assess because of the naturally broad range in body mass of individuals even within litters. While I'm not suggesting the conclusions are wrong, I wonder if there is a better way to display the data? Instead of normalising the mass of KO pups to that of WT mean/litter could the distribution of body mass be shown for both KO and WT, which would show the shift to the left for KO body mass and also how the body mass range overlaps.

There is a further anomaly in the description of body mass when describing that all organs were uniformly affected, "such that body proportion was maintained" (Lines 186-187). Crucially, no skeletal muscles (nor BAT which becomes important later) were included in Figure S3B, yet the DEXA data (Figure 2 G-I) clearly shows a reduction in lean mass of Zdbf2-KO animals, but not adipose mass. The DEXA data therefore establishes that body proportions have not been maintained.

Together with the tendency to gain less weight than their WT counterparts immediately after birth the Zdbf2-KO pups also exhibit reduced survival. The data look convincing but the description could be clearer. I suggest avoiding use of the terms "failure to thrive" and "survivability". To firm things up, it would be useful to put p values on the numbers of animals alive at key time-points using Chi-squared tests. The comparison of mixed WT+KO litters versus KO-only litters to assess the impact of competition on KO pup survival is a nice experiment but the conclusion "More likely, the impaired survival of Zdbf2-KO neonates is a consequence of intra-litter competition with healthier WT pups." could perhaps be tweaked. It may be better to say, e.g. "perinatal mortality was rescued when there is no competition with WT littermates." Importantly, it is not made clear whether KO pups in the all-KO litters exhibited a similar growth-deficient phenotype seen in KO pups from mixed litters. If so, then the growth deficient phenotype is not based on competition with WT littermates and removing the competition simply allows for more KO offspring to survive the critical first day or so after birth.

The importance of Zdbf2 dosage is next tested using a hypomorphic knock-down allele, which shows a less severe post-natal growth phenotype. Then a GOF allele is used to show that increasing Zdbf2 expression through (incomplete) activation of the normally silent maternal allele results ins early post-natal overgrowth. These are nice experiments that take the work beyond the all-or-nothing nature of most KO v WT comparisons. The results are not clear-cut since only male Zdbf2-GOF animals exhibit enhanced weight gain and statistical analysis of data could be more robust, as detailed below. As for the previous analysis of Zdbf2-KO animals, organ weight data is interpreted to show that all organs are equally affected (Figure S5D), however it is again a striking omission that no skeletal muscles were included among the organs dissected and weighed. Moreover, in this case there is no DEXA data to show how lean and adipose mass contribute to the increased total body mass. Finally, the importance of Zdbf2 dosage is reinforced through an "allelic switching" experiment.

Further experiments show that at a gross level the growth phenotype of Zdbf2-KO animals is not obviously due to gross changes in hypothalamus or pituitary development or in the ability to express various pituitary hormones. Circulating growth hormone levels are also found to be approximately normal at PN5 and PN15, leading the authors to conclude that GH involvement in the phenotype can be excluded. This is perhaps a little overstated because: (1) A transient deficiency in GH closer to the time of birth has not formally been excluded, which is important as the weight data suggest the growth deficiency occurs prior to PN5; (2) Altered sensitivity to GH has not been excluded. In contrast, evidence is shown for an IGF1 deficiency that provides a plausible molecular basis for the growth phenotype. The authors don't formally test the role of IGF1 (which would likely require a lot of work, e.g. to perform some form of rescue experiment) but the deficiency leads them to investigate growth rate using the weight data presented earlier identify (Figure 5E). The differences between WT and KO animals are subtle but most likely correct, in that they provide a better description of the data than that provided earlier. The conclusions would be more robust if the data were subject to some form of statistical testing. Overall though, a nice link is made between the reduction in Igf1 and poor feeding immediately post-partum which ultimately makes sense in terms of the expression pattern of Zdbf2 in brain and pituitary. Organ weights are revisited again, this time in PN3 pups. BAT was included for the first time among organs weighed and found to be the only organ reduced in mass in KO pups. It is admittedly difficult to measure skeletal muscle mass at this young stage but a histological assessment of muscle size would be possible and potentially informative. A reduction in mass of skeletal muscle and WAT, both derivatives of My5+ mesodermal tissue, as a consequence of the IGF1 deficiency immediately after birth, could explain the weight and body composition data presented in the paper. It would be helpful if more data could be included on these two tissues.

Transcriptomic data are presented consistent with transient disruption to feeding circuit genes in the hypothalamus of Zdbf2-KO pups with Npy and Agrp among only eleven genes mis-regulated at PN3. This provides further support for the mechanism underlying Zdbf2 function. It would be nice to see these data validated and extended. For instance, it could be argued that by PN3 the feeding deficiency has already been established and, therefore, interesting to establish whether expression of Nyp and Agrp is disrupted earlier (e.g. by qRT-PCR at late fetal to PN1 stages).

Specific points:

Lines 60-61. The paper by Andergassen et al., (2017: *eLife*, 6:e25125. DOI: 10.7554/*eLife*.25125) would be a good addition here.

Lines 130-143. I note the detailed description of Zdbf2 expression in hypothalamic structures is consistent with analysis in a paper currently on bioRxiv (preprint doi: https://doi.org/10.1101/2020.07.27.222893) that they may wish to cite.

Line 174 (FiguresS2A,B). Data look OK but would be useful to see some Chi-square tests done on the data to back up the statement that at birth mendelian and sex ratios were normal.

Line 175-176. "Failure to thrive" is a loose term here that does not accurately describe the data in Figures2B-D, which show a deficit in the ability KO pups to gain weight, in comparison with WT litermates. Appropriate stats are needed (ANOVA, not t-tests).

Line 177. For "2 weeks of age" refer to "PN15". Use days in this pre-weaning period to match the figure (2C).

Lines 178-179. Change "Body weight reduction persisted through adulthood". Body mass measurements are shown up to 12 weeks of age, which is only a fraction of adult life.

Lines 182-3. The statement, "not due to a developmental delay" seems a little too emphatic given the level of analysis is to assess gross morphological features. Perhaps e.g. "not obviously"?

Line 185 (Figure S2F-I). How many animals scored in each case?

Line 186. Change "all organs" to e.g. "a range of organs" since not all organs of the body have been sampled.

Figure S3B shows organ weight proportional to body mass, which is fine but it would also be useful to show a direct comparison of organ mass for WT and KO animals.

Lines 189-200. A number of figure references are incorrect, perhaps due to a late decision to remove or move one or more panels.

Lines 206-207. The inability to detect ZBDF2 protein is a weakness (we are all at the mercy of good antibodies) but it is nice to see this weakness discussed, here and elsewhere. Overall, the weight of evidence for the Zdbf2-KO being a null allele is strong.

Line 209. Again "fail to thrive" is a loose and unhelpful term that could be replaced with something more descriptive.

Lines 216-220. Several more figure references are incorrect. Also, is there a Table missing for Survival of pups carrying the maternal Liz deletion?

Lines 243-244. The statement "Interestingly, these mice displayed significant weight reduction compared to their WT littermates (Figure S4D)" needs to be backed up through appropriate statistical testing (e.g. ANOVA) of the weight data.

Line 245-246. Describe the data more accurately. Presumably "on average" refers to a comparison of means and consider making a more direct comparison with the Zdbf2-KO data.

Line 248. The end of this sentence needs rewriting to make proper sense.

Line 251. "The Zdbf2-GOF lines carry a ~ 900bp" is inaccurate as overlapping deletions of 924 and 768 bp are shown in S4E. Also, it seems more important to point out that both deletions remove 5' portions of the gDMR region and Liz exon 1.

Line 255. This sentence conflates sDMR methylation assays (Figure 3A and B) with RT-qPCR expression analysis (Figure 3C). All needs to be made clearer.

Figure 3C. Stats should be ANOVA, not repeated t-tests.

Line 262. It is inaccurate to state "animals were consistently overweight" when only males appear to be affected. Also, the difference for males appears marginal (Figure 3D) and there are no stats to support it. It would help to reorder the description of Results, e.g. explain first that the difference is specific to males and peaks at around PN15 during the pre-weaning period (3D and E). Male Zdbf2-GOF animals then remain heavier than WT into adult life, measured until 10 weeks of age (3F). Stats at 2 wks of age (3G) support this.

Line 263, "viability was normal". More accurately, 'viability was similar to that of WT controls'.

Line 266-267, "increased body weight of Zdbf2-GOF males was more pronounced during the nursing period (Figure 3D, G and H). Is this true? The differences are subtle in the pre-weaning period, as shown in Figure 3C and although appearing to diminish after PN21 this effect is not evident in Figures3E and 3F.

Line 271, "while females displayed standard normal distribution". This could be made clearer, e.g. 'female Zdbf2-GOF animal had a weight distribution similar to that of WT'. Also need to describe the body mass distribution of males, where it appears the majority are larger than the WT mean but some are smaller (the distribution looks potentially bimodal).

Line 342. IGF1 is a regulator of post-natal growth, not just a marker (also, use commas not dashes to punctuate).

Lines 355-365. Analysis of growth rates. Care is needed in the interpretation, especially when considering the narrowing of the weight difference between WT and KO animals from 1 week to 6 weeks of age, which is associated with an "enhanced post-pubertal growth spurt". Is the % difference significant given that total body weight is considerably increased over this period?

Line 373, "Having determined that Zdbf2-KO animals exhibit an Igf-1-like phenotype". This could be better expressed, e.g. "Having determined that Zdbf2-KO animals are deficient in Igf-1".

Lines 396-409. Data in the two figures described in this section (Figures6 and S7C) need more careful explanation in order to be clearer about which genes are significantly misregulated in the RNA-seq data.

Line 408. Change "continuously misregulated in the hypothalamus at 10dpp" to "also misregulated in the hypothalamus at 10dpp".

Line 409. I dislike the use of the phrase "catch-up phase" for reasons outlined earlier (see comment on lines 355-365).

Line 489-490. It is not clear that you are able to distinguish support for the parental conflict hypothesis (Zdbf2 acting as a paternally expressed promoter of resource acquisition from the mother) from that for co-adaptation, based on your evidence. Support in favour of the letter would require demonstration of a reciprocal function in the mother, such as an influence on milk let-down or nurturing behaviour.

Line 499-504. In the case of PWS the hyperphagia is prolonged and results in excess adipose deposition for which you have no evidence in your GOF mice. The lack of excess body weight out to 18 months is not shown and it would be interesting to know if the reduction in lean body mass and BAT is maintained, which might impact the glucose handling capability of Zdbf2-KO mice.

*Reviewer #2:*

Glaser et al., generated loss- and gain-of-function mutations in Zdbf2, where the former reduced protein levels and the latter increased them. They carefully analyzed expression of Zdbf2 throughout the hypothalamus and pituitary. They carefully quantified the growth deficits (or increases) in both male and female mice. Although they sought to identify the underlying mechanism for the growth loss (or gain, in the case of the gain of function mutants), they largely found that most of the known regulators of appetite and growth were not changed. This negative data is important. Plasma IGF-1 was decreased in the Zdbf2 knockout mice, and Npy and Agrp seemed increased in RNAseq data from hypothalami. They conclude that perhaps the lack of nutrition in neonates led to decreased IGF-1.

Strengths:

– The generation of the mutant mice was elegant, careful, and well-informed based on previous knowledge.

– The characterization of the mice was careful and performed with adequate numbers to give confidence in the findings.

– The inclusion of Zdbf2 loss of function and gain of function mutations demonstrates a role for the gene in regulating growth and provides excellent tool to better understand the mechanisms of how it impacts growth.

– The manuscript is well-written and easy to understand.

Weaknesses:

– The impact of this finding is predicated on the importance of imprinted genes in the hypothalamus and the hypothalamus in growth regulation. The authors cite Gregg et al., (2010) to justify the importance of imprinted genes in the hypothalamus, but that same paper also claimed that there were ~2,000 imprinted genes, a finding which has not been replicated.

– The lack of a similar role for ZDBF2 in human disease should be discussed carefully, because it points to the role of the gene/protein in placenta. ZDBF2 is expressed (and likely imprinted, according to publicly available GTex data) in many different human tissues. The relevance to humans should be discussed because it has broader implications for the conserved function of imprinted genes. In this case, the placenta seems to be the primary driver.

– It is helpful to compare this mouse model to the PWS mouse, because the PWS mouse model also shows failure to suckle that can be rescued by removing wild-type litter mates and reduced growth. However, the authors refer to PWS and then discuss the Magel2 deficient mouse, due to the metabolic defects. The PWS IC deletion mouse does not have clear metabolic defects (at least to my knowledge!) and MAGEL2-deficient humans do not phenocopy PWS. The authors should more carefully discuss this literature in the introduction and discussion.

– The poor suckle in the neonates precedes reduced milk and reduced growth. It seems pretty hard to disentangle cause and effect. Do pups with wild-type litter mates end up smaller than pups with only ko litter mates? Are the IGF-1 levels improved in mice with only ko litter mates compared to those with wt litter mates? This would go a long way to supporting the idea that IGF-1 levels follow after reduced milk intake.

– In Figure 6a-the authors should include pups from the ko/ko matings to compare 3 dpp milk weight in pups that only have ko competition. When wt litter mates are included, many of the kos are destined to die, and most of the death appears to occur in the first week. Are pups close to death driving this difference?

– It would be important to learn whether plasma IGF-1 levels catch up after 15 dpp. Perhaps a more careful time course of IGF-1 levels in the various genotypes and littermate competition situations should be followed, if this is the hypothesis as to why the mice are smaller (and larger in GOF Zdbf2 mutants; see next comment).

– Given that the GOF Zdbf2 mice show increased body weight, the authors should measure plasma IGF-1 to see if it is increased. It may help disentangle cause and effect of feeding and IGF-1. Do GOF Zdbf2 male mice have increased milk weight?

– The details of the RNA-seq experiments should be provided. How many replicates of each genotype were used? What sequencing depth? How many significantly DE genes in total and do they reveal anything when looking at the data in total (i.e. gene ontology)? It is not clear how solid the NPY and AGRP data are without this information.

Altogether, the data strongly support a role for Zdbf2 in regulating growth in mice. However, the data do not reveal the mechanism underlying this. The reduced IGF-1 levels are correlated with the reduced growth. The hypothesis that this is driven by reduced neonatal feeding is not convincingly supported by the data. It is also not clear whether Zdbf2 performs a similar function in humans.

These data may be very helpful in understanding the relationship between feeding and growth in mouse models. This mouse looks very much like a PWS-IC deletion mouse model. Perhaps similar pathways are disrupted in these models, and may reveal mechanisms of neonatal failure to thrive-is it hypotonia or lack of drive to eat? On one hand, understanding this in mice is not terribly important, but due to similarity with PWS, it has implications for human disease and perhaps why that mouse model does not model all of the features of the human disorder.

– The IGF-1 levels should be more carefully measured over a developmental time course and in ko mice with and without wild-type litter mates.

– The 3 dpp milk/stomach weight should also be quantified in the ko mice with ko only litter mates.

– Both IGF-1 levels and 3 dpp stomach weights should be measured in GOF mice. This might reveal whether drive to eat is really the effect of Zdbf2 perturbation or whether the IGF-1 levels are affected independent of feeding.

– The details of the RNA-seq experiments should be provided. These should include how many replicates and the sequencing depth. The data should be deposited in a public database, as well.

– As discussed above, the role of imprinted genes in the hypothalamus should be discussed more carefully. This example is seemingly specific to the mouse gene. The human ZDBF2 gene is expressed in many tissues and only IUGR may be impacted by expression defects in humans, and this seems to be maternally-expressed ZDBF2 (?) in placenta. As written, it oversells the impact of the data here. Although ZDBF2 regulates growth in mice, it isn’t clear that this also happens in humans and it’s not clear the mechanism, although it is correlated with low IGF-1 levels.

– Specifically, the authors shouldn’t discount the Gregg et al., finding that there are many more imprinted genes and then cite the paper to justify the importance of imprinted genes in the hypothalamus.

– The Magel2 deficient mouse is not a PWS mouse model and humans with deletions of MAGEL2 do not have overt metabolic defects. The details of the PWS versus Magel2 mouse model should be articulated more accurately.

– The potential for divergence between this role for Zdbf2 in humans versus mouse should be discussed. Even if it is a mouse-specific phenotype, it might help understand how feeding/growth is regulated. It may be that mutations in Zdbf2 are missed in humans, but this possibility should be discussed.

*Reviewer #3:*

In this manuscript, Glaser et al., report further analysis of the physiological effects of imprinting of the Liz-Zdbf2 locus in the mouse. Liz-Zdbf2 is of particular interest, because its imprinting is conferred by a transient, maternal germline differentially methylated region (DMR) which orchestrates the long-term monoallelic expression of Zdbf2. In previous work, the authors showed that abrogation of Liz transcription prevents expression of Zdbf2, resulting in postnatal growth retardation (Greenberg et al., 2017); they also reported prominent expression of Zdbf2 in the hypothalamus and pituitary dependent on prior Liz transcription. In the current manuscript, the authors explore the role of hypothalamic Zdbf2 expression and dosage, and the resulting phenotypic correlates, in more detail.

First, the authors generate a loss-of-function mutant of Zdbf2 to validate that the growth phenotype found upon deletion of Liz and resultant inability to express Zdbf2 can definitively be attributed to Zdbf2. They then report a very detailed assessment of the growth phenotype, making important observations, such as it is restricted to early post-natal period, there is a fully proportionate effect on organ weights, there is reduced viability in the immediate post-natal period that can be mitigated by removal of competition from wild-type litter mates, etc. The authors then go on to show there is a dosage effect, which they are able to do by use of a hypomorphic allele and a serendipitously produced gain-of-function allele. Demonstrating that there is a phenotypic effect (modest over-growth) of near biallelic expression is an important outcome for an imprinted gene – rather few studies have done this. Further, by combining the various alleles they have in hand, the authors show that providing Zdbf2 expression from the maternal rather than paternal allele (albeit at reduced level) is equally functional (bar a slight reduction in weight), suggesting that which parental allele expression emanates from is immaterial. This finding leads to some limited discussion about the evolutionary basis for genomic imprinting.

Zdbf2 is expressed in hypothalamus structures and the anterior and intermediate pituitary, however, no detectable effect on development of these tissues, presence of the various hormone-producing cells in the pituitary, is apparent in the knock-out, although RNA-seq revealed transient reduction in transcript levels of the oxerogenic neuropeptides AgRP and NPY. Reduced IGF1 circulating levels, which could account for growth restriction, which is apparently not mediated by any effect on GH, lead the authors to conclude that IGF1 reduction may be secondary to reduced food intake in the immediate perinatal period.

The strengths of this manuscript are that the work is done to a very high standard: the analysis is detailed and comprehensive, the data look robust, and the results and their implications discussed in a highly informed and accessible manner.

A weakness is that the molecular function of ZDBF2 is left open.

In addition, in relation to the inferences they draw on the evolution of imprinting, the authors need to be cautious not to make over-interpretations and thereby risk misleading readers.

1. The molecular function of ZDBF2 is left open. By RNA-seq, the authors report down-regulation of transcripts for AgRP and NPY in early postnatal hypothalamus, but does this reflect a direct effect on the regulation of AgRP and NPY transcription or impaired development or maturation of AgRP/NPY neurons? Immunofluorescence studies should enable an estimate to be made of the AgRP/NPY neuron population to identify any deficit. The authors comment on the lack of ZDBF2 antibodies, which would preclude co-IF studies to evaluate whether ZDBF2 is expressed within and has a cell-autonomous effect in these neurons. Perhaps co-IF using a β-galactosidase antibody in the gene-trap line would be productive.

2. In relation to their experiment in which they 'reverse' the imprinting of Zdbf2 by enforcing expression from the maternal allele, the authors conclude that "Zdbf2 functions on postnatal body homeostasis in a dose-dependent but parent-of-origin independent manner" (line 449-450) and "the parental information is not essential per se." (line 454). The authors need to be cautious not to over-interpret this result, and thereby risk misleading readers, as it tests the status quo rather than evolutionary history of the allele (which is not possible to do). The parental origin is probably more a reflection of the evolution of the imprinted state. For example, population genetic based theories, such as the 'conflict hypothesis', predict invasion of growth-promoting paternal alleles in the population in the context of maternal restraint. But at this point in time, the expression has attained an optimum so that artificially switching allelic expression whilst largely retaining dosage is predicted to be inconsequential (all other things being equal). Once the allele is expressed in offspring, its parental origin is not apparent other than in relation to other imprinted genes in cis.

3. Further in the Discussion (line 488-490), the authors conclude that "This function agrees with the co-adaptation theory according to which genomic imprinting evolved to coordinate interactions between the offspring and the mother." The authors should also consider that their results are equally compatible with the kinship/parental conflict theory. Indeed, you see this in the observation that Zdbf2-KO offspring survive better when there is no competition from wild-type littermates, i.e., a stronger Zdbf2 allele (the WT) that is better able to extract resources from mothers would spread more quickly through the population than the weaker allele (the KO). This is completely aligned with a kinship/conflict model.

4. In the Discussion (line 465-467), the authors comment that "similar IGF-1-related growth defects have been reported upon alteration of other imprinted loci: the Dlk1-Dio3 cluster and the Cdkn1c gene." The authors should consider adding Rasgrf1, whose role in IGF-1 regulation of postnatal growth was reported earlier (Itier et al., 1998), and which probably represents the first imprinted gene implicated in post-natal growth control. Rasgrf1 might also be useful as a counterpoint in the discussion as that report and subsequent studies (e.g., Drake et al., 2009) implicated defects in the hypothalamic-pituitary axis and growth hormone regulation.

---

## [Author Response]

Reviewer #1:[…]The authors establish that mouse Zdbf2 expression is restricted to hypothalamus and pituitary using both data generated themselves and that from publically available transcriptomic datasets (Figures1 and S1). However, none of the analysis rules out expression of Zdbf2 elsewhere during fetal or early post-natal life and I wonder if this has been established elsewhere?

From emouseatlas (http://www.emouseatlas.org/emagewebapp/pages/emage_data_browse.jsf), *Zdbf2* appears expressed in most brain structures (including pituitary gland) at E14.5; some expression is also detected in spinal cord, upper leg and upper arm.

We also used our *Zdbf2*-LacZ reporter line to assess *Zdbf2* expression in the developing embryo. Β-galactosidase staining at E12.5 embryonic days supports expression of *Zdbf2* in the developing brain and in the spinal cord, as depicted now as Figure 1—figure supplement 1B.

It is important because expression in, for example, skeletal muscle could provide an alternative explanation for the described phenotype. Also, they show levels of expression in both hypothalamus and pituitary arising during the pre-weaning period, between PN4 and PN21, then remaining high well into adulthood (PN70) (Figure 1E,F). Given that the major effect of Zdbf2 KO on growth is later shown to occur immediately after birth it would strengthen the paper to show expression during this period (PN1-3) and earlier, in the fetal brain.

We have now included 1dpp as an earlier timepoint in the *Zdbf2* RT-qPCR of hypothalamus and pituitary gland, as reported in Figure 1E-F. Expression levels are similar to 4dpp, confirming that noticeable increase occurs past 7dpp. Due to the difficulty to collect hypothalamic and pituitary glands from fetal brains (specifically and in adequate quantity), we did not investigate prenatal timepoints. Moreover, no difference in weight has been detected before birth (Figure 2B). Finally, regarding the comment about skeletal muscle, we did not see any expression in this tissue in our multi-tissue RT-qPCR assessment (Figure 1—figure supplement 1A, quadriceps and tongue).

In addition, they might comment in the Discussion on the persistent and relatively high level of expression seen in adults, which points to further functions of Zdbf2 in later life, perhaps relating to behaviour.

Thanks for the comment, we have adjusted the discussion (last sentence of the first Discussion paragraph). (Lines 436-437)

Zdbf2 KO pups are shown to have a deficit in their ability to gain weight in the immediate post-natal period, in comparison with WT littermates (Figure 2). My major concern here is the statistical treatment of the data: No test appears to have been applied to data in Figure 2B and for that in Figures2C and 2D the use of a Mann Whitney t-test, presumably applied repeatedly to pairs of samples at each time-point, is inappropriate. Instead, ANOVA or similar should be used.

As recommended, we have improved our statistical tests. Figures 2B, Figure 3—figure supplement 1D and Figure 3D now present a comparison of each data point with a Mann Whitney t-test. Here, we are comparing two groups (WT versus KO) independently at each time point, a Mann Whitney t-test is thus the appropriate test to use. Regarding the growth curves in Figures 2C-D and 3E-F, we implemented a two-way ANOVA test comparing WT and KO for all time points, globally. Results of the test are depicted in the graphs and explained in the legends.

Also, the description of the data is inaccurate: "Growth reduction was the most drastic prior to weaning, from 5 to 21dpp" (Lines 176-177). In fact, KO pups gain less weight than their WT littermates during the period PN1-PN5. It would be better to describe the differences in mass between WT and KO animals that the graphs show, rather than growth which is inferred (and more properly analysed later on in Figure 5E). It is potentially confusing to talk about a "growth reduction" in KO pups because the mass of both WT and KO animals increases over time (i.e. both are growing but the KOs less than normal). It would be more accurate to say, e.g. "while there is essentially no difference in mass at e18.5, KOs appear smaller than WT littermates at PN1 and PN5, indicating slower growth in the first few days after birth. The lower mass of KO animals persists into adulthood, measured up to 12 weeks of age". Here, I am also taking issue with the statement that the body weight reduction had "a tendency to minimize with age (Figure 2B)" (Line 179) because any difference in Figure 2B is subtle and the body mass difference is still evident post weaning (Figure 2C) and is not obviously reduced.

Thanks for the remark. These clarifications have been incorporated in the text.

I would be more cautious with the statement, "The reduced body weight phenotype was fully penetrant (Figure S2E)." First, the figure legend says 'highly' penetrant, which seems a better term to use because there are animals with mass equal to or greater than mean WT mass. Also, penetrance is difficult to assess because of the naturally broad range in body mass of individuals even within litters. While I'm not suggesting the conclusions are wrong, I wonder if there is a better way to display the data? Instead of normalising the mass of KO pups to that of WT mean/litter could the distribution of body mass be shown for both KO and WT, which would show the shift to the left for KO body mass and also how the body mass range overlaps.

We have taken this suggestion into account and replaced the graph in Figure 2—figure supplement 1E by a curve showing body mass distribution for both WT and KO at 15dpp, instead of normalization over WT. We can clearly see the shift of the distribution of KO mass to the left. The same has been applied for the *Zdbf2*-GOF body mass distribution in males, where a body mass shift towards bigger animals is observed (Figure 3—figure supplement 2E). We have also replaced “fully” by “highly” penetrant in the main text.

There is a further anomaly in the description of body mass when describing that all organs were uniformly affected, "such that body proportion was maintained" (Lines 186-187). Crucially, no skeletal muscles (nor BAT which becomes important later) were included in Figure S3B, yet the DEXA data (Figure 2 G-I) clearly shows a reduction in lean mass of Zdbf2-KO animals, but not adipose mass. The DEXA data therefore establishes that body proportions have not been maintained.

We corrected this statement accordingly.

Together with the tendency to gain less weight than their WT counterparts immediately after birth the Zdbf2-KO pups also exhibit reduced survival. The data look convincing but the description could be clearer. I suggest avoiding use of the terms "failure to thrive" and "survivability". To firm things up, it would be useful to put p values on the numbers of animals alive at key time-points using Chi-squared tests.

The terms “failure to thrive" and "survivability" have been corrected and a Chi2 test has been applied to the last data point of both viability graphs (Figure 2J-K).

The comparison of mixed WT+KO litters versus KO-only litters to assess the impact of competition on KO pup survival is a nice experiment but the conclusion "More likely, the impaired survival of Zdbf2-KO neonates is a consequence of intra-litter competition with healthier WT pups." could perhaps be tweaked. It may be better to say, e.g. "perinatal mortality was rescued when there is no competition with WT littermates."

We corrected this statement accordingly. (Lines 227-228)

Importantly, it is not made clear whether KO pups in the all-KO litters exhibited a similar growth-deficient phenotype seen in KO pups from mixed litters. If so, then the growth deficient phenotype is not based on competition with WT littermates and removing the competition simply allows for more KO offspring to survive the critical first day or so after birth.

We have ourselves thought about this at large. However, as pointed above by this reviewer, “body mass of individuals shows a naturally broad range, even within litters”. As there is no WT littermates in this experimental set up, we would need here to make comparison between litters, on such an intrinsically variable measurable as the weight, and which happens to be affected in a rather subtle manner in the *Zdbf2*-KO animals. Three types of crosses would be required: KO-only, KO from mixed KO and WT pups and also, WT-only. Inter-litter variability could be modulated by keeping only litters with similar sizes, and by redoing the three types of crosses at the same time, to reduce seasonal impact. This would involve a high number of crosses and mice, and still a significant amount of uncertainty regarding our ability to make conclusions, as inter-litter differences will still persist. We therefore decided to not do these measurements and we hope that the reviewer will understand.

The importance of Zdbf2 dosage is next tested using a hypomorphic knock-down allele, which shows a less severe post-natal growth phenotype. Then a GOF allele is used to show that increasing Zdbf2 expression through (incomplete) activation of the normally silent maternal allele results ins early post-natal overgrowth. These are nice experiments that take the work beyond the all-or-nothing nature of most KO v WT comparisons. The results are not clear-cut since only male Zdbf2-GOF animals exhibit enhanced weight gain and statistical analysis of data could be more robust, as detailed below. As for the previous analysis of Zdbf2-KO animals, organ weight data is interpreted to show that all organs are equally affected (Figure S5D), however it is again a striking omission that no skeletal muscles were included among the organs dissected and weighed. Moreover, in this case there is no DEXA data to show how lean and adipose mass contribute to the increased total body mass. Finally, the importance of Zdbf2 dosage is reinforced through an "allelic switching" experiment.

Thanks for the comments. The main text has been modified regarding the conclusion on the organ weight data and the same statistical tests were implemented as for the *Zdbf2*-KO data.

Further experiments show that at a gross level the growth phenotype of Zfb2-KO animals is not obviously due to gross changes in hypothalamus or pituitary development or in the ability to express various pituitary hormones. Circulating growth hormone levels are also found to be approximately normal at PN5 and PN15, leading the authors to conclude that GH involvement in the phenotype can be excluded. This is perhaps a little overstated because: (1) A transient deficiency in GH closer to the time of birth has not formally been excluded, which is important as the weight data suggest the growth deficiency occurs prior to PN5; (2) Altered sensitivity to GH has not been excluded.

Thanks for this remark. We removed this strong conclusion from the results and have added some nuance in the discussion (Line 471-472).

In contrast, evidence is shown for an IGF1 deficiency that provides a plausible molecular basis for the growth phenotype. The authors don't formally test the role of IGF1 (which would likely require a lot of work, e.g. to perform some form of rescue experiment) but the deficiency leads them to investigate growth rate using the weight data presented earlier identify (Figure 5E). The differences between WT and KO animals are subtle but most likely correct, in that they provide a better description of the data than that provided earlier. The conclusions would be more robust if the data were subject to some form of statistical testing.

It is unclear to us what kind of statistical test could be performed here. Charalambous 2012 uses the same type of growth rate graph and did not apply any test (DOI 10.1016/j.cmet.2012.01.006). Each data point was generated from this formula: (average weight at day *n* – average weight at day *n-1*)/ average weight at day *n-1*. Statistics have already been done on weight data, when looking at the growth curves (Figures 2C and D). What we have done though is to plot, as a matter of comparison, the growth rate curve from animals from the maternal transmission crosses (now as Figure 5—figure supplement 2A), who are asymptomatic (no weight difference). Maternal KO growth rate appears similar to WT, unlike what we observed in paternal KO animals. Although no statistics were applied here, this reinforces that the difference we see in the *Zdbf2*-KO growth rate curve is real.

Overall though, a nice link is made between the reduction in Igf1 and poor feeding immediately post-partum which ultimately makes sense in terms of the expression pattern of Zdbf2 in brain and pituitary. Organ weights are revisited again, this time in PN3 pups. BAT was included for the first time among organs weighed and found to be the only organ reduced in mass in KO pups. It is admittedly difficult to measure skeletal muscle mass at this young stage but a histological assessment of muscle size would be possible and potentially informative. A reduction in mass of skeletal muscle and WAT, both derivatives of My5+ mesodermal tissue, as a consequence of the IGF1 deficiency immediately after birth, could explain the weight and body composition data presented in the paper. It would be helpful if more data could be included on these two tissues.

This could be interesting investigating indeed, and could give some future work to be done with laboratories that are specialized in muscle development (which is well beyond our own expertise).

Transcriptomic data are presented consistent with transient disruption to feeding circuit genes in the hypothalamus of Zdbf2-KO pups with Npy and Agrp among only eleven genes mis-regulated at PN3. This provides further support for the mechanism underlying Zdbf2 function. It would be nice to see these data validated and extended. For instance, it could be argued that by PN3 the feeding deficiency has already been established and, therefore, interesting to establish whether expression of Nyp and Agrp is disrupted earlier (e.g. by qRT-PCR at late fetal to PN1 stages).

We have now performed RT-qPCR analysis of *Npy* and *Agrp* expression in hypothalamus at 1 and 3dpp. First, at 1dpp, we found that mRNA levels of both genes are already significantly downregulated in Zdbf2-KO (Figure 6F). This result is to be considered with our new measurements of plasmatic IGF-1 levels in 1dpp animals (as requested by Reviewer 2): at birth, IGF-1 is already reduced, although not as importantly as at 5 or 15dpp (Figure 5C). Second, at 3dpp, still by RT-qPCR, we found significant decrease of *Agrp* expression (downregulated but not significantly in RNA-seq) but a non-significant decrease of *Npy* expression (significantly downregulated in RNA-seq). Importantly, *Npy* and *Agrp* were no longer significantly downregulated later on, from our RNA-seq data at 10dpp (Figure 6—figure supplement 1D-E). Altogether, RNA-seq and RT-qPCR data support downregulation of two major hypothalamic genes controlling feeding behavior immediately at birth and during the first days of post-natal life. Additionally, as suggested by Reviewer 3, we have now performed Immunofluorescence on brain sections of *Zdbf2*-LacZ animals at 15dpp. We were thus able to validate that *Zdbf2* is co-expressed with NPY in hypothalamic cells.

Specific points:

All specific points have been addressed. We sincerely thank Reviewer 1 for dedicating so much effort correcting our manuscript and making some really helpful suggestions to improve its quality and accuracy.

Lines 60-61. The paper by Andergassen et al., (2017: eLife, 6:e25125. DOI: 10.7554/eLife.25125) would be a good addition here.

Added

Lines 130-143. I note the detailed description of Zdbf2 expression in hypothalamic structures is consistent with analysis in a paper currently on bioRxiv (preprint doi: https://doi.org/10.1101/2020.07.27.222893) that they may wish to cite.

Added

Line 174 (FiguresS2A,B). Data look OK but would be useful to see some Chi-square tests done on the data to back up the statement that at birth mendelian and sex ratios were normal.

Done

Line 175-176. "Failure to thrive" is a loose term here that does not accurately describe the data in Figures2B-D, which show a deficit in the ability KO pups to gain weight, in comparison with WT litermates. Appropriate stats are needed (ANOVA, not t-tests).

Done

Line 177. For "2 weeks of age" refer to "PN15". Use days in this pre-weaning period to match the figure (2C).

Done

Lines 178-179. Change "Body weight reduction persisted through adulthood". Body mass measurements are shown up to 12 weeks of age, which is only a fraction of adult life.

Changed

Lines 182-3. The statement, "not due to a developmental delay" seems a little too emphatic given the level of analysis is to assess gross morphological features. Perhaps e.g. "not obviously"?

Corrected

Line 185 (Figure S2F-I). How many animals scored in each case?

Appears now in the legend of the Figures (Figure 2—figure supplement 1F-H) (*n*=18 WT and *n*=10 KO)

Line 186. Change "all organs" to e.g. "a range of organs" since not all organs of the body have been sampled.

Done

Figure S3B shows organ weight proportional to body mass, which is fine but it would also be useful to show a direct comparison of organ mass for WT and KO animals.

A direct comparison of organ mass for WT and KO was performed but this graph, to our opinion, does not add information compared to the graph showing organ mass proportional to body mass in Figure 2—figure supplement 2B. You can see this graph with the raw data of organ mass measurement in Author response image 1.

**Author response image 1. sa2fig1:** Comparison of organ mass for wild-type and *Zdbf2*-KO males at 6 weeks-old.

Lines 189-200. A number of figure references are incorrect, perhaps due to a late decision to remove or move one or more panels.

Corrected

Lines 206-207. The inability to detect ZBDF2 protein is a weakness (we are all at the mercy of good antibodies) but it is nice to see this weakness discussed, here and elsewhere. Overall, the weight of evidence for the Zdbf2-KO being a null allele is strong.

Thanks

Line 209. Again "fail to thrive" is a loose and unhelpful term that could be replaced with something more descriptive.

Done

Lines 216-220. Several more figure references are incorrect. Also, is there a Table missing for Survival of pups carrying the maternal Liz deletion?

Corrected and two tables have been added (Figure 2- source data1B and D).

Lines 243-244. The statement "Interestingly, these mice displayed significant weight reduction compared to their WT littermates (Figure S4D)" needs to be backed up through appropriate statistical testing (e.g. ANOVA) of the weight data.

As for Figure 2B, data represent the percentage of mutant weight, normalized to their WT littermates. Normalization has been done per litter to avoid weight difference resulting from difference in litter size. We applied at Mann-Whitney t-test comparing two groups (WT versus mutant) at each time point and show significant results on the graph (Figure 3—figure supplement 1D).

Line 245-246. Describe the data more accurately. Presumably "on average" refers to a comparison of means and consider making a more direct comparison with the Zdbf2-KO data.

Done

Line 248. The end of this sentence needs rewriting to make proper sense.

Done

Line 251. "The Zdbf2-GOF lines carry a ~ 900bp" is inaccurate as overlapping deletions of 924 and 768 bp are shown in S4E. Also, it seems more important to point out that both deletions remove 5' portions of the gDMR region and Liz exon 1.

Corrected

Line 255. This sentence conflates sDMR methylation assays (Figure 3A and B) with RT-qPCR expression analysis (Figure 3C). All needs to be made clearer.

Done

Figure 3C. Stats should be ANOVA, not repeated t-tests.

Done

Line 262. It is inaccurate to state "animals were consistently overweight" when only males appear to be affected. Also, the difference for males appears marginal (Figure 3D) and there are no stats to support it. It would help to reorder the description of Results, e.g. explain first that the difference is specific to males and peaks at around PN15 during the pre-weaning period (3D and E). Male Zdbf2-GOF animals then remain heavier than WT into adult life, measured until 10 weeks of age (3F). Stats at 2 wks of age (3G) support this.

Done, including new statistics.

Line 263, "viability was normal". More accurately, 'viability was similar to that of WT controls'.

Corrected

Line 266-267, "increased body weight of Zdbf2-GOF males was more pronounced during the nursing period (Figure 3D, G and H). Is this true? The differences are subtle in the pre-weaning period, as shown in Figure 3C and although appearing to diminish after PN21 this effect is not evident in Figures3E and 3F.

We have added statistics on Figure 3D showing that the gain in body mass appears almost significant at 5dpp in Zdbf2-GOF males (p-value 0.055) and significant from 7 to 21dpp, using a Mann-Whitney t-test. On that graph, the increase is still visible at 28dpp but does not appears as significant. The growth curve from Figure 3E measuring weight differences from 1 to 19dpp (nursing period, prior to weaning) shows more significant difference between WT and GOF than the measurements from 3 to 10 weeks on Figure 3F (after weaning). Altogether, those observations indicate that the increased of body weight in this mutant is more pronounced during the nursing period.

Line 271, "while females displayed standard normal distribution". This could be made clearer, e.g. 'female Zdbf2-GOF animal had a weight distribution similar to that of WT'. Also need to describe the body mass distribution of males, where it appears the majority are larger than the WT mean but some are smaller (the distribution looks potentially bimodal).

We have changed the type of graph, as suggested previously for similar type of data in *Zdbf2*-KO pups, by a curve showing body mass distribution for both WT and KO at 15dpp (Figure 3—figure supplement 2E). With this new graph, the bimodal distribution is less visible as before with the normalization towards WT weights, suggesting that most GOF pups were larger than WT controls, as written in the main text (Line 272-273).

Line 342. IGF1 is a regulator of post-natal growth, not just a marker (also, use commas not dashes to punctuate).

Modified

Lines 355-365. Analysis of growth rates. Care is needed in the interpretation, especially when considering the narrowing of the weight difference between WT and KO animals from 1 week to 6 weeks of age, which is associated with an "enhanced post-pubertal growth spurt". Is the % difference significant given that total body weight is considerably increased over this period?

As mentioned previously, it is unclear to us what kind of statistical test could be performed on this growth rate curve.

Line 373, "Having determined that Zdbf2-KO animals exhibit an Igf-1-like phenotype". This could be better expressed, e.g. "Having determined that Zdbf2-KO animals are deficient in Igf-1".

Corrected

Lines 396-409. Data in the two figures described in this section (Figures6 and S7C) need more careful explanation in order to be clearer about which genes are significantly misregulated in the RNA-seq data.

Better description provided. We also now provide Figure 6- source data 1 with all misregulated genes and their RPKM, logFC and FDR values in the RNA-seq data at both 3 and 10dpp.

Line 408. Change "continuously misregulated in the hypothalamus at 10dpp" to "also misregulated in the hypothalamus at 10dpp".

Done

Line 409. I dislike the use of the phrase "catch-up phase" for reasons outlined earlier (see comment on lines 355-365).

Removed

Line 489-490. It is not clear that you are able to distinguish support for the parental conflict hypothesis (Zdbf2 acting as a paternally expressed promoter of resource acquisition from the mother) from that for co-adaptation, based on your evidence. Support in favour of the letter would require demonstration of a reciprocal function in the mother, such as an influence on milk let-down or nurturing behaviour.

The discussion has been modulated.

Line 499-504. In the case of PWS the hyperphagia is prolonged and results in excess adipose deposition for which you have no evidence in your GOF mice. The lack of excess body weight out to 18 months is not shown and it would be interesting to know if the reduction in lean body mass and BAT is maintained, which might impact the glucose handling capability of Zdbf2-KO mice.

Thanks for the comment. We did not keep *Zdbf2*-GOF animals for ageing experiments. However, measurements were made for *Zdbf2*-KO animals, which kept on being smaller/lighter than WT littermates at 18 months. Although our cohort is of small number, results were consistent. We have added “data not shown” in this sentence.

Reviewer #2:[…]– The IGF-1 levels should be more carefully measured over a developmental time course and in ko mice with and without wild-type litter mates.

We have added measurements of circulating IGF-1 at 1dpp, thereby showing now a time course from 1 to 15dpp (see new panel Figure 5C). Interestingly, we observed decreased IGF1 in *Zdbf2*-KO pups at 1dpp already, which then becomes more and more pronounced over time, reaching 70% decrease at 15dpp. Importantly, we also now report that *Npy* and *Agrp* downregulation (which could relate to poor suckling in mutant neonates, and hence reduced IGF-1 levels) is also noticeable by RT-qPCR as early as 1dpp (see new panel Figure 6F). Including this 1dpp point was therefore crucial, and we thank this reviewer for the suggestion. However, we did not explore IGF-1 circulating levels at timepoint later than 15dpp: body mass reduction in the mutants does not become more important with age (Figure 2B-D), and *Npy* and *Agrp* resume to normal levels at 10dpp (Figure 6—figure supplement 1E).

Regarding measurements in *Zdbf2*-KO-only pups, because of the lack of WT littermates, we would need here to make comparison between litters, on such an intrinsically variable as plasma IGF-1 levels. We would actually need three types of crosses: KO-only, KO from mixed KO and WT pups and also, WT-only. Inter-litter variability could be modulated by keeping only litters with similar sizes, and by redoing the three types of crosses at the same time, to reduce seasonal impact. This would involve a high number of crosses and mice, and still a significant amount of uncertainty regarding our ability to make conclusions, as inter-litter differences will still persist. We therefore decided to not do these measurements and hope that the reviewer will understand.

– The 3 dpp milk/stomach weight should also be quantified in the ko mice with ko only litter mates.

We did not perform this experiment for the reason stated above. The subtle phenotype and lack of littermate’s comparison will render the conclusions of those experiments difficult to interpret.

– Both IGF-1 levels and 3 dpp stomach weights should be measured in GOF mice. This might reveal whether drive to eat is really the effect of Zdbf2 perturbation or whether the IGF-1 levels are affected independent of feeding.

We have now measured stomach weights at 3dpp and IGF-1 plasma levels at 5 and 15dpp in *Zdbf2*-GOF males and WT littermates. Regarding stomach weight (Figure 6—figure supplement 1A), no significant difference was observed at 3dpp between GOF and WT. This goes along with the fact that we did not observe a significant body mass increase in GOF males at this same age (Figures 3D). On Author response image 2 provided hereafter to the reviewer only, we can appreciate that *Zdbf2*-KO males at 3dpp are smaller than WT littermates and for most of them, this is correlated with smaller stomach mass. However, there is no such correlation for the *Zdbf2*-GOF males. This implies that increased body mass might not result from enhanced feeding after birth, but rather from later signaling leading to increased body mass visible at 15 dpp (Figure 3G-H and Figure 3—figure supplement 2E).

**Author response image 2. sa2fig2:** Correlation between body mass and stomach mass at 3dpp for *Zdbf2*-KO males (left panel) and *Zdbf2*-GOF males (right panel).

Regarding IGF-1 (Figure 5—figure supplement 2B), plasma levels were also not different in *Zdbf2*-GOF compared to WT males, at both 5dpp and 15dpp. IGF-1 may not be the main driver of the male-specific increased body weight upon ZDBF2 overexpression. Other hypothalamic circuits might be affected. Alternatively, IGF-1 might be (only transiently) affected in these GOF mutants but we were not able to detect it with our experimental set up. We mention these possibilities in the Results.

– The details of the RNA-seq experiments should be provided. These should include how many replicates and the sequencing depth. The data should be deposited in a public database, as well.

Sorry for the omission, *n*=3 replicates were used per genotype at 3dpp, and *n*=2 at 10dpp. This information has been added in the legend and in the method section, where we also inform the reader about the sequencing depth. Data have been publicly deposited under the GEO number GSE153265, as it was originally reported in the Material and Method section (under “Data Availability”). Regarding the comment about the number of significant DE genes and their GO analysis: as reported in the results, there are 11 differentially expressed (DE) genes in *Zdbf2*-KO hypothalami at 3dpp compared to WT littermates. The list of these DE genes was provided in Figure 6—figure supplement 1E, as well as the list of DE genes at 10dpp (*n*=16). We have now worked on being clearer in the text regarding the RNAseq results. Also, raw values of the RNA-seq are now provided as Figure 6- source data 1. Regarding GO analysis, this could not be performed on such small gene set.

– As discussed above, the role of imprinted genes in the hypothalamus should be discussed more carefully. This example is seemingly specific to the mouse gene. The human ZDBF2 gene is expressed in many tissues and only IUGR may be impacted by expression defects in humans, and this seems to be maternally-expressed ZDBF2 (?) in placenta. As written, it oversells the impact of the data here. Although ZDBF2 regulates growth in mice, it isn't clear that this also happens in humans and it's not clear the mechanism, although it is correlated with low IGF-1 levels.

Thanks for opening this discussion. As in the mouse, ZDBF2 is highly expressed in brain tissues in humans, including the pituitary gland (GTEXdata). It has also been shown to be expressed in human placenta (where it is also paternally expressed, as shown by our own work in Duffié et al., Genes and Dev 2014 and by others, such as Monteagudo-Sanchez et al., Clinical Epigenetics 2019). It is difficult to know whether placental expression is relevant, as there is no direct comparison with brain tissues in terms of levels. In the mouse, we can score modest expression in the placenta (Figure 1—figure supplement 1A)—which we previously showed to emanate from the labyrinth layer in Greenberg, Glaser et al., Nature Genetics 2017—but the level is by 10-fold lower than in the hypothalamus, and we did not find any placental phenotype upon *Zdbf2* depletion in this previous study. Concerning ZDBF2 role in human trait and physiology, there has indeed been an association between IUGR and reduced *ZDBF2* expression. However, difference was only seen in male conceptuses with IUGR, not females, and notably, ZDBF2 levels in the placenta of IUGR males were similar to the levels seen in the placenta of control female conceptuses. It is also unclear as to whether the reduction in expression is cell autonomous, there could be differences in the cellular composition of IUGR placentae compared to controls. Finally, it could also be that the IUGR condition triggers ZDBF2 downregulation, rather than the reverse. As a whole, it is true that there is not so much literature about ZDBF2 in humans and that is mostly linked to placenta, but we feel that the link between ZDBF2 and placental function in humans is far from being proved, and would require independent cohorts to be analyzed, single-cell analysis to avoid confounding effects etc….

Nonetheless, we have now nuanced the discussion, as we cannot indeed exclude that ZDBF2 may perform different functions in mice and humans : “Finally, decreased *ZDBF2* levels have recently been associated with intra-uterine growth restriction (IUGR) in humans (Monteagudo-Sánchez *et al.,* 2019). Whether this is linked to impaired fetal IGF-1 production or to distinct roles of ZDBF2 related to placental development in humans would be interesting to assess, in regards to conservation or not of imprinted gene function across mammals” (Lines 484-488).

– Specifically, the authors shouldn't discount the Gregg et al., finding that there are many more imprinted genes and then cite the paper to justify the importance of imprinted genes in the hypothalamus.

We originally cited Gregg et al., (2010) for their analysis of expression of known (and well established) imprinted genes, showing their prevalent expression in the hypothalamus compared to other brain structures. To further justify that the hypothalamus is a privileged site for imprinted gene expression, we now refer to a more recent study (preprint) that explored brain scRNA-seq datasets and showed overrepresentation of (known) imprinted gene expression in hypothalamic neurons [Higgs et al., bioRxiv 2021 (https://doi.org/10.1101/2020.07.27.222893)].

– The Magel2 deficient mouse is not a PWS mouse model and humans with deletions of MAGEL2 do not have overt metabolic defects. The details of the PWS versus Magel2 mouse model should be articulated more accurately.

We are sorry if we were unclear. Our point is not to compare *Magel2*-KO mice with PWS patients (which has been done by others, including the refence we cite) but to emphasize that there are cases where initial poor feeding and altered capacity to gain weight after birth can be associated with obesity and metabolic defects later in life. We have modified our sentence: “This is observed in mouse KO models of the imprinted *Magel2* gene that map to the Prader Willi syndrome (PWS) region, *recapitulating some features of PWS patients,* who after a failure to thrive as young infants exhibit a catch-up phase leading to overweight and hyperphagia (Bischof *et al.,* 2007).” (Line 516-520).

– The potential for divergence between this role for Zdbf2 in humans versus mouse should be discussed. Even if it is a mouse-specific phenotype, it might help understand how feeding/growth is regulated. It may be that mutations in Zdbf2 are missed in humans, but this possibility should be discussed.

As mentioned in #5, this nuance has been added in the Discussion.

Reviewer #3:[…]1. The molecular function of ZDBF2 is left open. By RNA-seq, the authors report down-regulation of transcripts for AgRP and NPY in early postnatal hypothalamus, but does this reflect a direct effect on the regulation of AgRP and NPY transcription or impaired development or maturation of AgRP/NPY neurons? Immunofluorescence studies should enable an estimate to be made of the AgRP/NPY neuron population to identify any deficit. The authors comment on the lack of ZDBF2 antibodies, which would preclude co-IF studies to evaluate whether ZDBF2 is expressed within and has a cell-autonomous effect in these neurons. Perhaps co-IF using a β-galactosidase antibody in the gene-trap line would be productive.

Thanks for this suggestion. We have now performed co-localization analysis of NPY and ZDBF2 using anti-NPY and anti-β-galactosidase antibodies on 15dpp brain sections of *Zdbf2*-lacZ animals. We observed both signals in the paraventricular nucleus of the hypothalamus, the known site of NPY expression (see the new Figure 6D). This suggests that *Zdbf2* is expressed in NPY+ neurons.

We tried to confirm these results with an anti-AgRP antibody (rabbit anti-AgRP, Phoenix Pharmaceuticals cat. No. H-003-57), quite popular in the literature, at least on adult brains. Unfortunately, we failed to obtain specific staining in pre-puberal brains after several round of troubleshooting.

We also tried to quantify NPY intensity and NPY-expressing cells on brain sections from WT and *Zdbf2*-KO at 3dpp but the results were not interpretable. First, NPY staining only allows detection of the fibers (local and projections) but not the neuron’s soma. It is extremely complicated to quantify the intensity of the signal in the fibers and impossible to count NPYexpressing cells. Second, we observed important variations in the intensity and distribution of the signal across different sections from the same animal, implying that comparison of WT and KO could only be done on sections from the exact same coordinate in the hypothalamus. Finally, the decrease of *Npy* mRNA from the RNA-data is rather mild, and its quantification by IF questionable. For these reasons, our attempt to visualize NPY downregulation and/or to count NPY+ cells using IF was, unfortunately, unsuccessful.

2. In relation to their experiment in which they ‘reverse’ the imprinting of Zdbf2 by enforcing expression from the maternal allele, the authors conclude that “Zdbf2 functions on postnatal body homeostasis in a dose-dependent but parent-of-origin independent manner” (line 449-450) and “the parental information is not essential per se.” (line 454). The authors need to be cautious not to over-interpret this result, and thereby risk misleading readers, as it tests the status quo rather than evolutionary history of the allele (which is not possible to do). The parental origin is probably more a reflection of the evolution of the imprinted state. For example, population genetic based theories, such as the ‘conflict hypothesis’, predict invasion of growth-promoting paternal alleles in the population in the context of maternal restraint. But at this point in time, the expression has attained an optimum so that artificially switching allelic expression whilst largely retaining dosage is predicted to be inconsequential (all other things being equal). Once the allele is expressed in offspring, its parental origin is not apparent other than in relation to other imprinted genes in cis.

We have tried to temper this sentence:

“The divergent DNA methylation patterns that are established in the oocyte and the spermatozoon provide opportunities to evolve mono-allelic regulation of expression, but once transmitted to the offspring, the parental origin of expression is not essential per se.” (Line 467-468).

3. Further in the Discussion (line 488-490), the authors conclude that “This function agrees with the co-adaptation theory according to which genomic imprinting evolved to coordinate interactions between the offspring and the mother.” The authors should also consider that their results are equally compatible with the kinship/parental conflict theory. Indeed, you see this in the observation that Zdbf2-KO offspring survive better when there is no competition from wild-type littermates, i.e., a stronger Zdbf2 allele (the WT) that is better able to extract resources from mothers would spread more quickly through the population than the weaker allele (the KO). This is completely aligned with a kinship/conflict model.

We have added this to the discussion, now considering both theories (Lines 508-510).

4. In the Discussion (line 465-467), the authors comment that “similar IGF-1-related growth defects have been reported upon alteration of other imprinted loci: the Dlk1-Dio3 cluster and the Cdkn1c gene.” The authors should consider adding Rasgrf1, whose role in IGF-1 regulation of postnatal growth was reported earlier (Itier et al., 1998), and which probably represents the first imprinted gene implicated in post-natal growth control. Rasgrf1 might also be useful as a counterpoint in the discussion as that report and subsequent studies (e.g., Drake et al., 2009) implicated defects in the hypothalamic-pituitary axis and growth hormone regulation.

Thanks for the remark. We have now added *Rasgrf1* in the discussion (Lines 482-485).